

# Stratospheric ozone trends and attribution over 1984-2020 using ordinary and regularised multivariate regression models

Yajuan Li[1,2], Sandip S. Dhomse[3,4], Martyn P. Chipperfield[3,4], Wuhu Feng[3,5], Jianchun Bian[2,6,7], Yuan Xia[1] and Dong Guo[8]

5  1 School of Electronic Engineering, Nanjing Xiaozhuang University, Nanjing, China
2 Key Laboratory of Middle Atmosphere and Global Environment Observation, Institute of Atmospheric Physics, Chinese Academy of Sciences, Beijing, China
3 School of Earth and Environment, University of Leeds, Leeds, UK
4 National Centre for Earth Observation (NCEO), University of Leeds, Leeds, UK
10  5 National Centre for Atmospheric Science (NCAS), University of Leeds, Leeds, UK
6 College of Earth and Planetary Sciences, University of Chinese Academy of Sciences, Beijing, China
7 College of Atmospheric Sciences, Lanzhou University, Lanzhou, China
Key Laboratory of Meteorological Disaster, Ministry of Education/Joint International Research Laboratory of Climate and Environment Change/Collaborative Innovation Center on Forecast and Evaluation of Meteorological Disasters, Nanjing
University of Information Science & Technology, Nanjing, China

*Correspondence to*: Yajuan Li (yajuanli@njxzc.edu.cn) and Sandip S. Dhomse (s.s.dhomse@leeds.ac.uk)

**Abstract**. Accurate quantification of long-term trends in stratospheric ozone can be challenging due to their sensitivity to
natural variability, the quality of the observational datasets, non-linear changes in forcing processes as well as the statistical methodologies. Multivariate linear regression (MLR) is the most commonly used tool for ozone trend analysis, however, the complex coupling in most atmospheric processes can make it prone to the over-fitting or multi-collinearity-related issues when using the conventional Ordinary Least Squares (OLS) setting. To overcome this issue, we adopt a regularised (Ridge) regression method to estimate ozone trends and quantify the influence of individual processes. Here, we use the Stratospheric
Water and OzOne Satellite Homogenized (SWOOSH) merged data set (v2.7) to derive stratospheric ozone profile trends for the period 1984-2020. Beside SWOOSH, we also analyse a machine-learning-based satellite-corrected gap-free global stratospheric ozone profile dataset from a chemical transport model (ML-TOMCAT), and output from two chemical transport model (TOMCAT) simulations forced with ECMWF reanalyses ERA-Interim and ERA5.

With Ridge regression, the stratospheric ozone profile trends from SWOOSH data show smaller declines during 1984-1997
compared to OLS with the largest differences in the lowermost stratosphere (>4 % per decade at 100 hPa). Upper stratospheric ozone has increased since 1998 with maximum (~2 % per decade near 2 hPa) in local winter for mid-latitudes. Negative trends with large uncertainties are observed in the lower stratosphere with the most pronounced in the tropics. The largest differences in post-1998 trend estimates between OLS and Ridge regression methods appear in the tropical lower stratosphere (with ~7 % per decade difference at 100 hPa). Ozone variations associated with natural processes such as the quasi-biennial oscillation
(QBO), the solar variability, the El Niño–Southern Oscillation (ENSO), the Arctic oscillation (AO) and the Antarctic oscillation (AAO) also indicate that Ridge regression coefficients are somewhat smaller and less variable compared to the OLS-based estimates. Additionally, ML-TOMCAT based trend estimates are consistent with SWOOSH data set. Finally, we argue that the large differences between the satellite-based data and model simulations confirm that there are still large uncertainties in ozone trend estimates especially in the lower stratosphere, and caution is needed when discussing results if
explanatory variables used are correlated.

## 1 Introduction

With the success of the Montreal Protocol and its amendments, the emission of ozone-depleting substances (ODSs) has been effectively banned and observations are able to detect decreases in their concentrations (e.g. Anderson et al., 2000; Solomon
et al., 2006; Chipperfield et al., 2017; Montzka et al., 2021), however quasi-global total column ozone does not show a



statistically significant ozone increase (WMO, 2022 and references therein). To some certain extent, there is a scientific consensus that the ODS-related positive ozone trends are balanced by the negative contributions from atmospheric dynamics (e.g., Weber et al., 2022; Bognar et al., 2022). As the impacts of chemical and dynamical processes on ozone variability are variable across the stratosphere, accurate quantification of stratospheric ozone trends remains an unresolved challenge.

An important aspect of long-term ozone trends that has been confirmed by various recent studies is that there is an ozone increase in the upper stratosphere (e.g. Harris et al., 2015; Chipperfield et al., 2017; Sofieva et al., 2017; Ball et al., 2017; Steinbrecht et al., 2017; Petropavlovskikh et al., 2019; Godin-Beekmann et al., 2022), partly due to the decreased ODS concentrations and partly due to the stratospheric cooling resulting from increased greenhouse gases (GHGs). However, our understanding about the evolution of lower stratospheric ozone remains highly uncertain. Various observation-based studies
suggest that there has been a continued decline in lower stratospheric ozone since 1998, both in the tropics and mid-latitudes (e.g. Ball et al., 2018; 2019a; Wargan et al., 2018; Orbe et al., 2020; Bognar et al., 2022), while model simulations do not reproduce these trends (Ball et al., 2020; Dietmüller et al., 2021; Li et al., 2022; Davis et al., 2022). It is well established that ozone in the lower stratosphere is sufficiently long-lived and primarily controlled by the transport and circulation changes (e.g. Chipperfield et al., 2018). The increasing GHGs induce a strengthening of tropical upwelling and enhance the stratospheric
circulation, which causes tropical ozone to decline in the lower stratosphere (Marsh et al., 2016). Besides, the non-linear quasi-biennial oscillation (QBO) and the El Niño–Southern Oscillation (ENSO) influence the dynamical variability in the lower stratosphere and drive the large interannual ozone variability in this region (Ball et al., 2019a; Diallo et al., 2018). The asymmetrical change pattern in the Brewer-Dobson circulation (BDC), with a relative slowdown in the northern hemisphere (NH), also provides evidence pointing to dynamically driven ozone variability in the lower stratosphere (e.g. Mahieu et al.,
2014; Stiller et al., 2017; Prignon et al., 2021; Bognar et al., 2022). Considering the inconsistencies between observations and model simulations, it is important to gain better insight about the causes of uncertainties in the estimates of the lower stratospheric ozone trends.

Most importantly, the quantification of stratospheric ozone trends is not only sensitive to the natural variability and non-linear forcing processes, it also depends on the quality of the observational datasets and the time periods considered. To determine
the long-term ozone trends and the attribution of ozone variability, composites of observations are generally used by merging different ozone observational data sets into a long, multi-decadal record. However, there are artefacts in the uncertainty budget and sampling inconsistencies between various datasets. Previous studies have used multiple composites merged from different observing platforms and discussed the sensitivity of ozone trends to the inclusion of new datasets (Ball et al., 2018, 2019; Sofieva et al., 2017, 2022; Steinbrecht et al., 2017; Petropavlovskikh et al., 2019; Weber et al., 2022; Godin-Beekmann et al.,
2022). Here, we use the merged Stratospheric Water and OzOne Satellite Homogenized (SWOOSH, version 2.7) data set to assess the stratospheric ozone trends (Davis et al., 2016) for the 1984-2020 time period. In addition, a machine-learning-based satellite-corrected gap-free global stratospheric ozone profile dataset from a chemical transport model (ML-TOMCAT, Dhomse et al., 2021) is also used for comparison.

To improve the assessment of the long-term ozone tends and variability, multivariate linear regression (MLR) models with
different configurations are most widely used by separating the influence of various chemical and dynamical processes on the ozone concentrations (e.g. Dhomse et al., 2006, 2022; Chehade et al., 2014; Li et al., 2020, 2022). Szeląg et al. (2020) analyzed the seasonal dependence of stratospheric ozone trends from four merged satellite datasets over 2000–2018 using a two-step MLR approach. Godin-Beekmann et al. (2022) presented the evaluation of stratospheric ozone profile trends in the extra-polar region over the period 2000–2020 with an updated version of the Long-term Ozone Trends and Uncertainties in the
Stratosphere (LOTUS) regression model which additionally included seasonal trend terms. Bognar et al. (2022) used both MLR and dynamical linear modelling (DLM) methods (Laine et al, 2014; Ball et al., 2017, 2019a) to determine the stratospheric ozone trends during 2000–2021 with a combination of three satellite datasets. Recently, Dhomse et al. (2022)



used an ensemble of MLR models and regularised regression methods (Ridge, Lasso and ElasticNet) to estimate the solar cycle signal in the observed and simulated ozone profiles for 2005-2020. With the extended datasets and improved statistical methodologies, there is better agreement and reduced uncertainties in different satellite-based ozone trends. However, it should be noted that trends in the lower stratosphere are still masked by large dynamical/natural variability.

Additional complications also arise from the use of chemical/dynamical proxies in the MLR, and some of them are inevitably correlated and coupled, causing an issue of over-fitting or multi-collinearity (e.g. Dhomse et al., 2022). Multi-collinearity refers to a condition in which some explanatory variables have a great influence on other explanatory variables, which will significantly lead to inconsistent and unreliable parameter estimates in regression modelling (e.g. Shariff and Duzan, 2018). To overcome multi-collinearity problem, regularised regression models such Ridge regression are highly recommended (e.g. Hoerl and Kennard, 1970). Previous studies have indicated that Ridge regression performs better than other estimators and can produce reliable results when multi-collinearity exists in the data (e.g. Shariff and Duzan, 2018; Tirink et al., 2020; Gana, 2022). In this paper, we use MLR models based on both OLS and Ridge regression methods to compare and discuss their differences in estimating stratospheric ozone trends. Besides SWOOSH and ML-TOMCAT data sets, two chemical transport model (TOMCAT) simulations forced with the European Centre for Medium-Range Weather Forecasts (ECMWF) reanalyses ERA-Interim and ERA5 (Li et al., 2022) are also used for comparison with satellite-based ozone trends and ozone changes associated with natural variability.

The paper is organized as follows. Section 2 describes the merged satellite-based ozone data set (SWOOSH), two TOMCAT model simulations forced with ECMWF reanalyses ERA-Interim and ERA5 (A_ERAI and B_ERA5), and a machine-learning-based satellite-corrected TOMCAT product (ML-TOMCAT). Section 3 describes the MLR models and regression methods based on OLS and Ridge. In Section 4 results regarding the ozone profile trends based on OLS and Ridge regression methods and the ozone variations associated with natural processes are presented. Our conclusions are summarized in Section 5.

## 2 Data

### 2.1 SWOOSH

The Stratospheric Water and OzOne Satellite Homogenized (SWOOSH) data set is a monthly mean record of stratospheric ozone and water vapor data from a subset of limb sounding and solar occultation satellites operating from 1984 to present (Davis et al., 2016). It is obtained from https://csl.noaa.gov/groups/csl8/swoosh/ (last access: Jan 2023). The SWOOSH (v2.7) record is comprised of several individual satellite data from the Stratospheric Aerosol and Gas Experiment (SAGE-II/III v7/v4), the Upper Atmospheric Research Satellite Halogen Occultation Experiment (UARS HALOE v19), the UARS Microwave Limb Sounder (MLS v5/6), the Aura MLS (v5), the Aura High Resolution Dynamics Limb Sounder (HIRDLS v7) and the Atmospheric Chemistry Experiment Fourier Transform Spectrometer (ACE-FTS v3.6) instruments, as well as a combined data product. The corrections that vary with latitude and height are determined from coincident observations closely matched in space and time during time periods of instrument overlap. The primary SWOOSH product consists of zonal-mean values at grids of 2.5, 5 and 10° resolution. There are filled and unfilled versions of the data set on both geographical and equivalent latitude coordinates. Many previous studies have demonstrated the reliability of this product in analyzing the variability and mechanisms associated with stratospheric ozone (e.g. Lu et al., 2019; Shangguan et al., 2019; Zhang et al., 2021; Hu et al., 2022). Here we use the gap-filled SWOOSH data at grids of 2.5° and 12 levels per decade ranging from 316 to 1 hPa (31 pressure levels). This SWOOSH data is considered a beta product and will continue to be updated as long as new data are available from the Aura MLS instrument or a suitable replacement.



## 2.2 TOMCAT simulations

Chemical transport models (CTMs) are important tools for understanding past ozone changes by combining up-to-date knowledge about various physical and chemical processes with a mathematically consistent framework. TOMCAT/SLIMCAT (hereafter TOMCAT) is a global 3-D off-line CTM (Chipperfield, 2006), which contains a detailed description of stratospheric

chemistry (e.g. Feng et al., 2011, 2021; Dhomse et al., 2015, 2016; Chipperfield et al., 2018) or tropospheric chemistry (Monks et al., 2017) and uses winds and temperatures from meteorological analyses (usually ECMWF) to specify the atmospheric transport and temperatures.

Here we have performed two TOMCAT simulations, A_ERAI and B_ERA5, which are forced with ECMWF ERA-Interim (Dee et al., 2011) and ERA5 (Hersbach et al., 2020) reanalysis datasets (e.g. Dhomse et al., 2019; Feng et al., 2021; Li et al.,

2022), respectively. As ERA5 has been released by ECMWF to supersede ERA-Interim which covers from January 1979 to August 2019, there are more and newer observations assimilated in ERA5. The inhomogeneities in reanalysis data sets could introduce spurious transport features (e.g. Schoeberl et al., 2003; Ploeger et al., 2015), and thus cause inability of chemical models to simulate the observed stratospheric ozone changes (Li et al., 2022). The two TOMCAT simulations are identical to those used in Li et al. (2022), with $2.8° \times 2.8°$ (T42 Gaussian grid) horizontal resolution and 32 hybrid sigma-pressure levels

ranging from the surface to about 60 km. The 6-hourly grid point meteorological fields are interpolated linearly in time for both runs.

## 2.3 ML-TOMCAT

We use a machine-learning-based method and chemically self-consistent output from a 3-D chemical transport model (CTM) to create a satellite-corrected long-term stratospheric ozone profile data set (ML-TOMCAT, Dhomse et al., 2021a). The CTM

(TOMCAT) setup is described in the following Sect. 2.2. A random-forest (RF) regression model, including five terms: passive ozone ($O_3$), HCl mixing ratio (HCl), methane mixing ratio ($CH_4$), Mg II solar flux term (MgII) as well as observation–model total column ozone difference (dTCO), is applied to the observation–model ozone difference by selecting 20 years of UARS-MLS (1991–1998) and AURA-MLS (2005–2016) measurements as a training period. The passive $O_3$, HCl and $CH_4$ are tracers taken from TOMCAT output fields, dTCO is calculated from Copernicus Climate Change Service (C3S) total ozone data, and

MgII is obtained from http://www.iup.uni-bremen.de/UUVSAT/Datasets/mgii (last access: Jan 2023). These variables account for possible biases in CTM profiles due to transport, solar flux variability or the use of coarse spectral bins (e.g. Dhomse et al., 2013; Sukhodolov et al., 2016; Feng et al., 2021).

The results show that ML-TOMCAT ozone concentrations are in excellent agreement with SWOOSH data and they are well within uncertainties of the observational data sets at almost all stratospheric levels. ML-TOMCAT is also ideally suited for

the evaluation of chemical model ozone profiles and observation-based data sets from the tropopause up to 0.1 hPa. The ML-TOMCAT ozone profile data (v1.0) on pressure and altitude levels in mixing ratios and number density units are available via https://doi.org/10.5281/zenodo.5651194 (Dhomse et al., 2021b).

## 3 Methods

### 3.1 Multivariate linear regression models

Here we use multivariate linear regression (MLR) models to estimate the stratospheric ozone trends and to separate the influence of important chemical and dynamical processes on the ozone variations. The MLR setup is a modified version from that used in Dhomse et al. (2022). Briefly, it has 77 terms, including 24 monthly linear trend terms and 24 intercept terms for the independent linear trends (ILT, e.g. Weber et al., 2018) before and after the turnaround year (1997) close to the timing of the peak stratospheric halogen loading, 24 QBO terms at 30 and 50 hPa, and 5 proxies for the 11-year solar cycle, El-Nino

Southern Oscillation (ENSO), Arctic Oscillation (AO), Antarctic Oscillation (AAO) and Eliassen-Palm (EP) flux (integrated over previous and current months). QBO, ENSO, AO and AAO indices are from Climate Prediction Center



([https://www.cpc.ncep.noaa.gov/](https://www.cpc.ncep.noaa.gov/), last access: Jan 2023). The proxy for EP flux uses the 50 hPa vertical component (Fz50) with 2-month mean values averaged over mid-latitudes between 45° and 75° in each hemisphere from the ECMWF ERA5 reanalysis. The effects of the aerosol loading from volcanic eruptions are not considered in the MLR as we remove data for 1991 and 1992.

We apply the MLR to monthly mean ozone anomalies and get

$$dO_3(t) = \sum_{j=1}^{77} \beta_j \times P_j(t) + \epsilon(t)$$

where $dO_3(t)$ denotes monthly mean ozone anomaly time series from 1984-2020 (1984-2018 for the simulation A_ERAI) obtained by referencing the monthly mean $O_3(t)$ to the climatological mean for each calendar month. The explanatory proxies $P_j$ include 77 terms which are de-trended (except for the linear trend terms) and normalised between 0 and 1. The coefficients $\beta_j$ are obtained by least squares fitting of the residuals.

As noted earlier, as most atmospheric processes are not completely independent, the MLR models suffer from multi-collinearity issues to a certain extent. Here we use both ordinary least squares (OLS) and regularised (Ridge) linear regression models for comparison to quantify the estimated ozone trends and the influence of individual processes.

**3.2 OLS regression**

Ordinary least squares (OLS) regression is a common method used to study the relationship between explanatory variables and response variables in regression models. The OLS method aims to minimize the sum of squared errors (SSE) between the observed values ($y_i$) and predicted values ($\hat{y}_i$). The objective function being minimized is written as

$$\text{minimize} \left( \text{SSE} = \sum_{i=1}^{n} (y_i - \hat{y}_i)^2 \right)$$

It should be noted that the OLS with unbiased estimators performs well only when all key regression assumptions are satisfied, e.g. linear relationship, more observations ($n$) than features ($p$), no or little multi-collinearity among the explanatory variables. With the presence of multi-collinearity, OLS will be not robust and will result in inaccurate model. Additionally, the OLS model is designed to minimise the residual errors but with relatively high variance, which means small changes in explanatory variables can lead to large changes in the estimated regression coefficients. Thus we should be careful of the results of parameter estimates and inference under the OLS procedure.

**3.3 Ridge regression**

To overcome the multi-collinearity issue in regression, a number of methods have been developed and the most common is Ridge regression (Hoerl and Kennard, 1970). Ridge regression is a type of regularized regression which gives a penalty (called an L2 penalty) as in Hastie et al. (2009) and Kuhn and Johnson (2013) to constrain the magnitudes and fluctuations of the coefficient estimates. This constraint helps to reduce the variance of the model at the expense of no longer being unbiased, which is a reasonable compromise. The objective function with a penalty term is written as

$$\text{minimize} \left( \text{SSE} + \alpha \sum_{j=1}^{p} \beta_j{}^2 \right)$$

The penalty is calculated as the square of the magnitude of coefficients. By adding this penalty term, all coefficients of the regression variables ($\beta_j$) will be constrained or shrunk, but not to zero, so they all remain in the model. The strength of the penalty term is controlled by a tuning parameter ($\alpha$). When this tuning parameter is set to zero, Ridge regression equals OLS regression. If $\alpha = \infty$, all coefficients in the regression are shrunk to zero. The ideal penalty is therefore somewhere in between 0 and $\infty$ that helps to control the model from over-fitting or under-fitting to the training data. Here we use cross-validation (CV) to identify the optimal $\alpha$ value when the mean square error (MSE) reaches the minimum (Pedregosa et al., 2011). The



Ridge regression model used here is from Python scikit module (For details see https://scikit-learn.org/stable/modules/linear_model.html, last access: Jan 2023).

**Figure 1** shows the SWOOSH ozone anomalies and fitting from OLS and Ridge regression models near the Equator (~1°N) at pressure levels of 1, 10 and 46.4 hPa. The cross-validated MSE and coefficients for the Ridge regression model are also shown as the α value grows from 0.01 to 100. In all cases shown in **Figure 1**, we find a slight improvement in the MSE as the penalty (α) gets larger, suggesting that a regular OLS model likely over-fits the training data. As the penalty continues to increase, coefficients in the Ridge regression model are shrunk until close to zero. The vertical dashed lines represent the optimal α value with the minimum MSE ($\alpha_0 = 0.174$, 0.048 and 0.026 in Ridge regression for ozone anomaly data at pressure levels of 1, 10 and 46.4 hPa). Monthly mean ozone anomalies as well as the OLS and Ridge fitting from ML-TOMCAT, simulations A_ERAI and B_ERA5 are shown in the supplement (**Figures S1-3**).

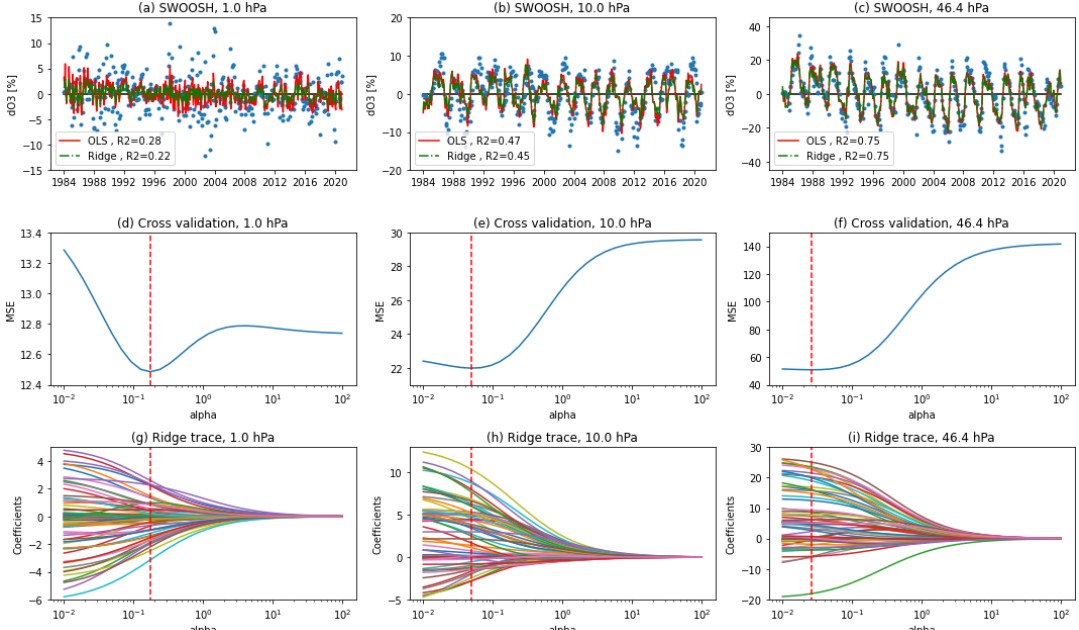

**Figure 1: (a-c)** Monthly mean ozone anomalies (blue dots) and the OLS (red line) and Ridge fitting (green dot-dashed line) from SWOOSH data during 1984-2020 at the pressure levels of 1 hPa (left column), 10 hPa (middle column) and 46.4 hPa (right column) for the 1°N latitude. **(d-f)** Cross-validated MSE values as well as **(g-i)** Ridge regression trace of the coefficients that change with alpha (α) are also shown. The vertical red dashed line indicates the optimal tuning value ($\alpha_0$) for Ridge regression where the minimal MSE exists.

## 4 Results and Discussion

### 4.1 Ozone profile trends with OLS and Ridge regression

**Figure 2** shows the annual mean stratospheric ozone profile trends (% per decade) compared between OLS and Ridge regression methods for three latitude bands (60-35°S, 20°S-20°N and 35-60°N) from SWOOSH, ML-TOMCAT and two model simulations (A_ERAI and B_ERA5) over the period 1984-1997. The trend results as well as the 2σ uncertainties (the standard deviation of the fit residuals) for several pressure levels (1, 2, 10, 46.4 and 100 hPa) are given in **Table 1**. With Ridge regression, the stratospheric ozone profile trends from SWOOSH data show smaller declines during 1984-1997 compared to OLS-based trend estimates. As shown in **Figure 2** and **Table 1**, large OLS-Ridge differences appear in the upper stratosphere



(>1 % per decade at 2 hPa) and the lowermost stratosphere (>4 % per decade at 100 hPa). With slightly larger uncertainties in
trend coefficients, the smaller negative trends in both the upper and lower stratosphere in Ridge regression indicate smaller
fluctuations in Ridge coefficients. Compared to the trends at mid-latitudes, there are larger decreases in the tropical lower
stratospheric ozone (-28 % per decade for OLS and -13 % per decade for Ridge regression) while they are insignificant with
large uncertainties (up to 23~24 % per decade). These large decreases and uncertainties to some extent are associated with the
considerable dynamical variability near the tropopause (e.g. Sofieva et al., 2014; Thompson et al., 2021; Bognar et al., 2022),
and also are related to the quality of the satellite data and limitations in sampling and resolution (Davis et al., 2016). Ridge-
based pre-1998 trend estimates from ML-TOMCAT and two model simulations show very good agreement with those from
SWOOSH data at mid-latitudes in both the Northern Hemisphere (NH) and Southern Hemisphere (SH). Large differences
appear in the tropical middle and lower stratosphere with a range of 2-4 % per decade trends near 30 hPa, and near-zero trends
in simulation A_ERAI near 100 hPa. We should note that there are large uncertainties in the lower stratosphere for both satellite
data and model simulations.

**Table 1: Stratospheric ozone trends with 2σ uncertainties (in % per decade) from SWOOSH during 1984-1997 based on OLS and Ridge regression.**

| Levels (hPa) | 60-35°S | | 20°S-20°N | | 35-60°N | |
|---|---|---|---|---|---|---|
| | OLS | Ridge | OLS | Ridge | OLS | Ridge |
| 1 | -3.39 (2.47) | -1.98 (2.56) | -1.36 (1.83) | -0.52 (1.93) | -5.46 (2.17) | -4.06 (2.22) |
| 2 | -5.94 (2.48) | -4.40 (2.56) | -4.07 (1.98) | -3.03 (2.02) | -6.96 (2.33) | -5.41 (2.38) |
| 10 | -0.42 (2.00) | -0.08 (2.16) | -0.47 (2.46) | -0.37 (2.56) | -2.86 (1.76) | -2.14 (1.82) |
| 46.4 | -3.74 (2.50) | -2.34 (2.65) | -3.68 (3.29) | -3.04 (3.45) | -2.52 (2.55) | -1.79 (2.62) |
| 100 | -8.18 (5.81) | -3.45 (6.17) | -28.05 (23.04) | -13.09 (24.54) | -10.94 (6.42) | -5.72 (6.89) |

**Table 2: Stratospheric ozone trends with 2σ uncertainties (in % per decade) from SWOOSH during 1998-2020 based on OLS and Ridge regression.**

| Levels (hPa) | 60-35°S | | 20°S-20°N | | 35-60°N | |
|---|---|---|---|---|---|---|
| | OLS | Ridge | OLS | Ridge | OLS | Ridge |
| 1 | 0.09 (1.19) | -0.17 (1.24) | -0.17 (0.88) | -0.13 (0.93) | 0.07 (1.04) | -0.13(1.07) |
| 2 | 1.58 (1.19) | 0.92 (1.23) | 1.57 (0.95) | 1.15 (0.97) | 1.69 (1.13) | 1.21 (1.15) |
| 10 | 0.59 (0.96) | 0.27 (1.04) | 0.22 (1.19) | 0.13 (1.23) | 0.23 (0.85) | 0.16 (0.88) |
| 46.4 | -0.19 (1.21) | -0.36 (1.28) | -1.78 (1.59) | -1.72 (1.66) | -0.49 (1.23) | -0.44 (1.26) |
| 100 | 0.57 (2.80) | -0.16 (2.97) | -14.65 (11.11) | -7.66 (11.84) | -3.76 (3.09) | -1.62 (3.32) |




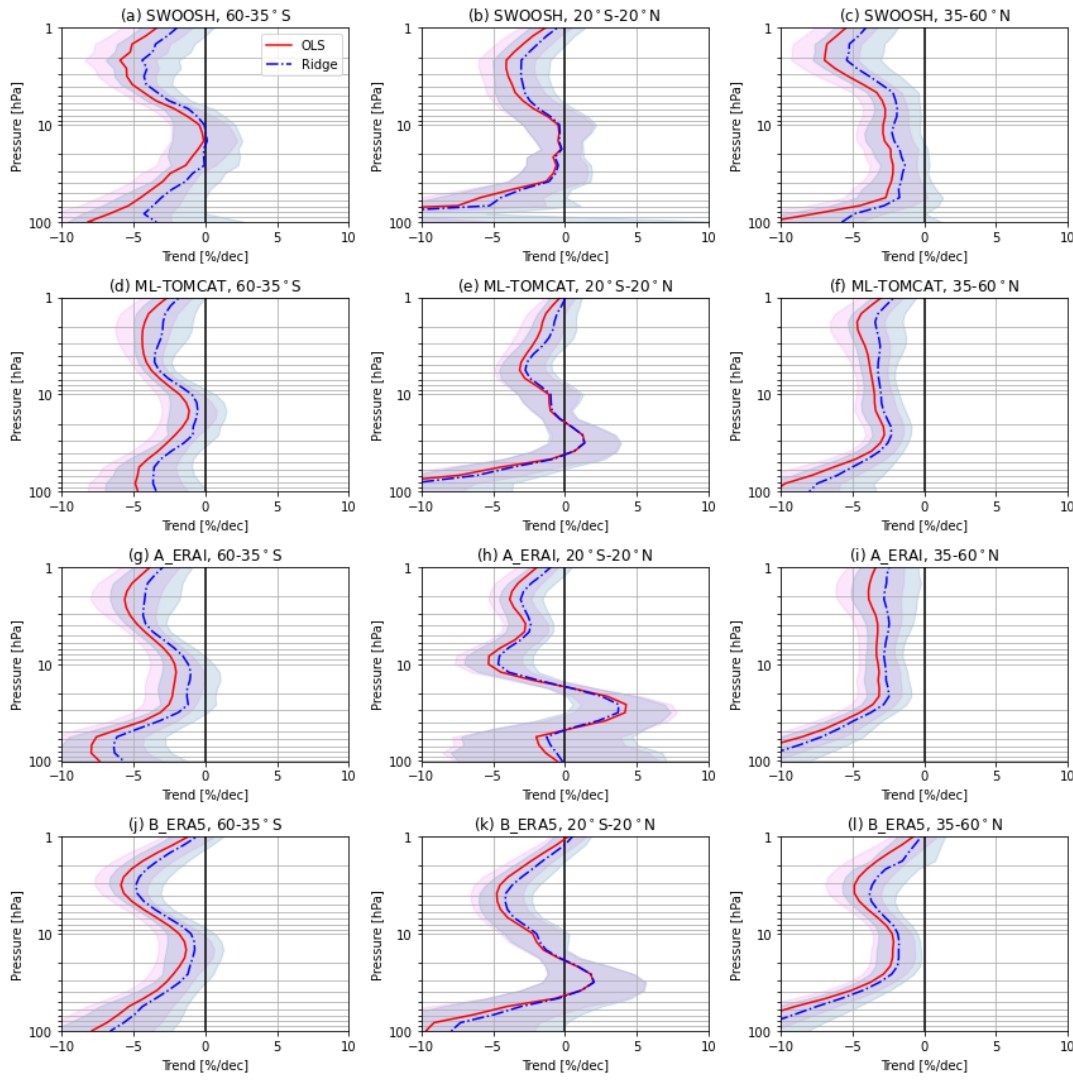

**Figure 2: Profiles of annual mean stratospheric ozone trends (% per decade) compared between OLS and Ridge regression methods**

**for three latitude bands (60-35°S, 20°S-20°N and 35-60°N) from (a-c) SWOOSH, (d-f) ML-TOMCAT, model simulations (g-i) A_ERAI and (j-l) B_ERA5 over the period 1984-1997. Shaded regions are 2-σ uncertainties.**





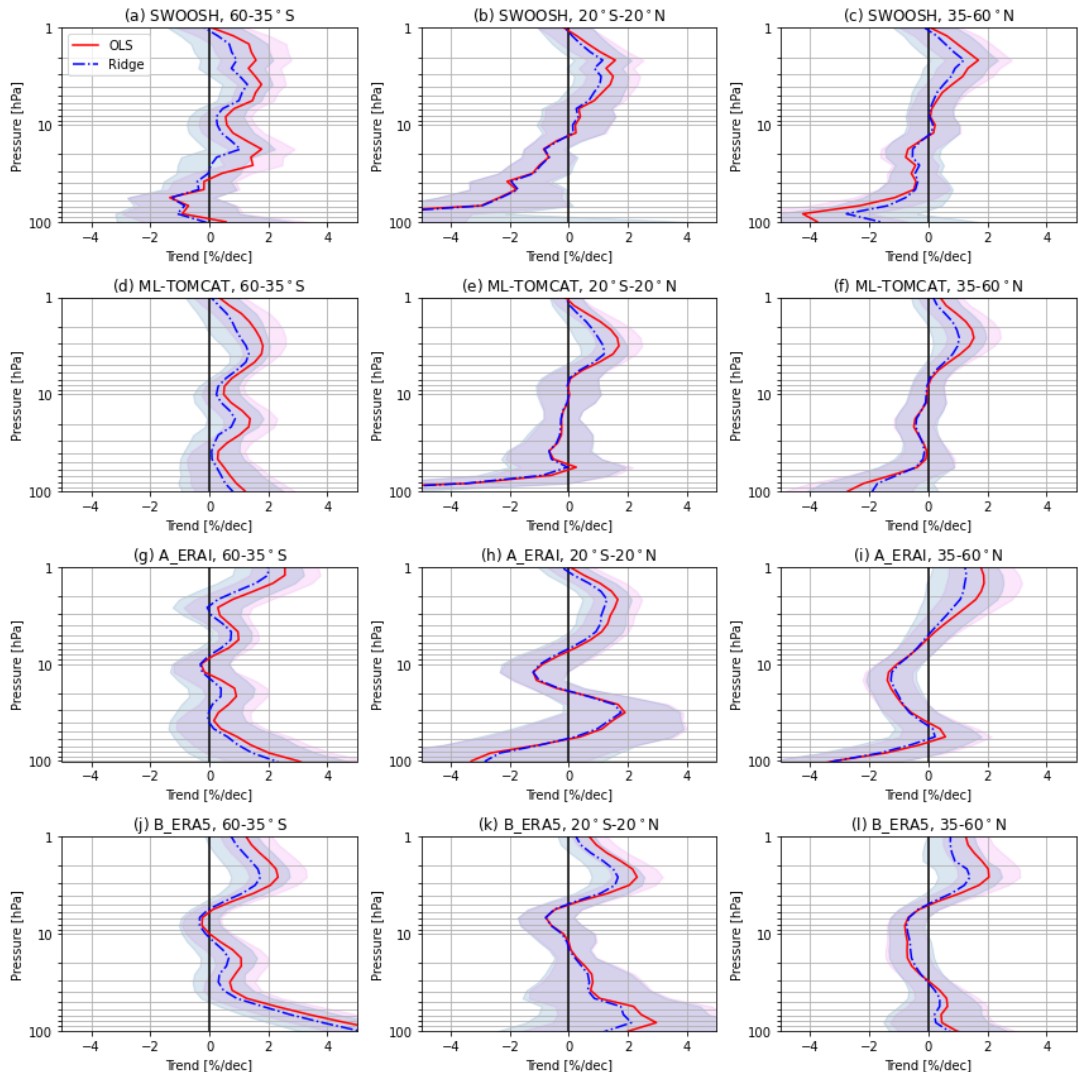

**Figure 3: Same as Figure 2 but for the post-1998 time periods (1998-2020 for SWOOSH, ML-TOMCAT and B_ERA5, and 1998-2018 for A_ERAI).**

As shown in **Figure 3**, upper stratospheric ozone has increased since 1998 across all the three latitude bands and the increases based on Ridge regression are relatively smaller. **Table 2** gives some trend results and corresponding 2-σ uncertainties from SWOOSH data during 1998-2020. The largest increase is 1.21±1.15 % per decade near 2 hPa in the NH mid-latitudes, 1.15±0.97 % per decade near 2 hPa in the tropics, and 1.77±0.85 % per decade at 3.8 hPa in the SH mid-latitudes, respectively. In the mid- and lower stratosphere, ozone trends are generally negative except for the non-significant positive trends near 20 hPa in the SH mid-latitudes where a large difference of ~1.4 % per decade occurs between OLS and Ridge regression methods. Negative trends with larger uncertainties are observed in the lower stratosphere and are most pronounced in the tropics (-7.66±11.84 % per decade at 100 hPa). The largest difference between OLS and Ridge regression methods occurs in the tropical lowermost stratosphere with a difference of ~7 % per decade at 100 hPa, followed by the NH mid-latitudes with >2 % per decade difference at 100 hPa.





Compared to the trend estimates from model simulations in **Figure 3**, the ML-TOMCAT data set shows more consistent results with those using SWOOSH data. Largest differences between SWOOSH- and ML-TOMCAT-based ozone trends appear in the SH mid-latitude lower stratosphere where ML-TOMCAT shows positive trends, and in the tropical mid- and lower

stratosphere with close to zero trends near 60 hPa (although these trends have large uncertainties). On the other hand, trends from both the model simulations (A_ERAI and B_ERA5) show largest inconsistencies with respect to SWOOSH-based trends in the lower stratosphere. Simulation B_ERA5 shows positive trends for all three latitude bands that are more pronounced in the SH mid-latitudes (5.05±2.04 % per decade at 100 hPa). Simulation A_ERAI also shows positive trends in the SH mid-latitudes where a large inconsistency also appears in the upper stratosphere with trends close to zero near 2 hPa. Although

simulation A_ERAI shows negative trends in both the tropical and NH mid-latitude lowermost stratosphere, there are large differences in the mid- and lower stratosphere with anomalous positive trends near 30 hPa (60 hPa) in the tropics (NH mid-latitudes). Overall, B_ERA5 shows better agreement with SWOOSH and ML-TOMCAT in the upper stratosphere, especially in the SH mid-latitudes, compared to A_ERAI. This simulated discrepancy might be associated with the uncertainties in temperature-dependent reaction rates in the models (e.g. Stolarski et al., 2010; Dhomse et al., 2013, 2016). While in the

lowermost stratosphere, both model simulations show very large discrepancies in ozone trends and B_ERA5 shows much larger and more overestimated positive trends for all latitudes. These differences between satellite-based datasets and model simulations suggest there are still large uncertainties in the lower stratosphere where dynamical processes dominate (Dietmüller et al., 2021; Li et al., 2022).

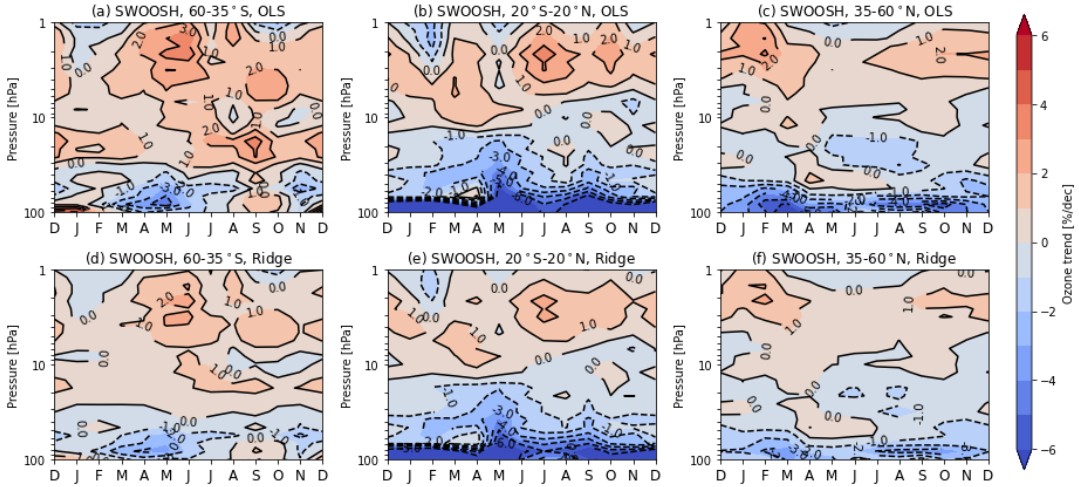


**Figure 4: Pressure-season variation of linear trends in ozone (% per decade) from SWOOSH data over 1998–2020 for three selected latitudinal bands (60-35°S, 20°S-20°N, 35-60°N) based on (a-c) OLS and (d-f) Ridge regression methods.**

The monthly mean variations of stratospheric ozone trends from SWOOSH data during 1998-2020 are averaged over three

latitude bands (60-35°S, 20°S-20°N, 35-60°N) and compared using both OLS and Ridge regression methods, as shown in **Figure 4**. There is a strong seasonal dependence in stratospheric ozone trends, with the signs of positive and negative trends varying with season and altitude. OLS-based trend estimates are in good agreement with those in previous studies (e.g. Szelag et al., 2020). Positive trends are observed in the upper stratosphere (10-1 hPa) for almost all seasons with the maximum (>2 % per decade) in local winter at mid-latitudes, while in the tropics (near 1-3 hPa) negative trends of more than -1% per decade

appear in December-January-February (DJF). In the middle stratosphere (32-10 hPa), there is a hemispheric asymmetric structure with positive trends (1-2 % per decade) in the SH mid-latitudes and negative trends (-1 % per decade) in the NH mid-latitudes in June-July-August (JJA). In the lower stratosphere (100-32 hPa), there are persistent negative trends for all seasons





in the tropics with the largest negative trends in May (< -4% per decade) and negligible trends in March and April near 60 hPa. Trends in the NH mid-latitudes are more negative in the lowermost stratosphere compared to those in the SH mid-latitudes. In

the SH mid-latitudes, there exists a clear transition from negative trends in February-July to positive trends in August-October. The Ridge regression method shows very similar results to those in OLS except that Ridge-based trends are constrained to some extent with smaller coefficients.

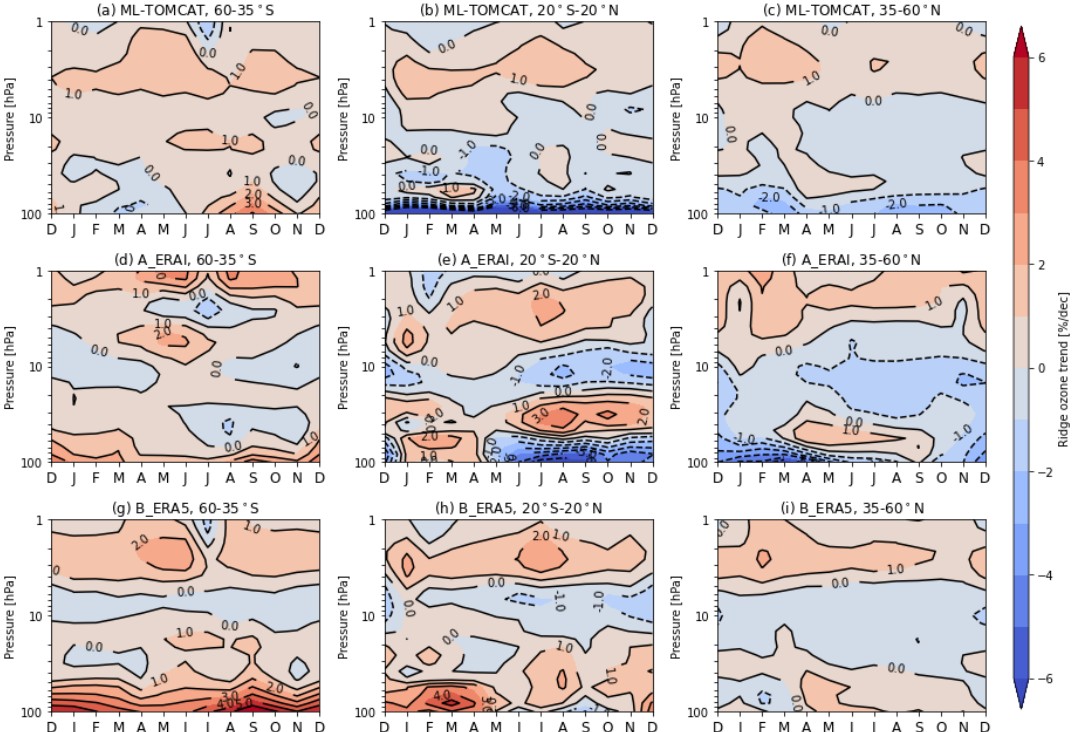

**Figure 5: Pressure-season variation of linear trends in ozone (% per decade) from (a-c) ML-TOMCAT, (d-f) A_ERAI and (g-i) B_ERA5 data over 1998–2020 (1998-2018 for A_ERAI) for three selected latitudinal bands (60-35°S, 20°S-20°N, 35-60°N) based on the Ridge regression method.**

**Figure 5** shows the comparison of seasonal variations of stratospheric ozone trends over the post-1998 period from ML-
TOMCAT data and two model simulations (A_ERAI and B_ERA5) based on Ridge regression. Trends from ML-TOMCAT data show more consistent seasonal dependence with those from SWOOSH data, while model-based estimates show significant differences. Simulation A_ERAI shows abnormal trend estimates in the SH upper stratosphere with minimal negative trends near 2-3 hPa in JJA (local winter), while both ML-TOMCAT and B_ERA5 data show positive trends. In the SH lowermost stratosphere, both model simulations show positive trends for all seasons, different from the trend pattern with seasonal
dependence from SWOOSH and ML-TOMCAT data. In the tropical mid- and lower stratosphere, there are large differences in seasonal ozone trends between model simulations and satellite data, and between the two model simulations. Simulation A_ERAI shows more negative trends (< -1 % per decade) near 10 hPa and more positive trends (>2 % per decade) near 30 hPa in June-December compared to ML-TOMCAT-based trends, which are consistent with the annual mean trends shown in **Figure 3**. Furthermore, simulation A_ERAI shows negative trends in the tropical lowermost stratosphere except for the
positive trends in January-April months. Compared to simulation A_ERAI, trends from B_ERA5 shows more positive trends for all seasons in the tropical lower stratosphere, opposite to the negative trends from SWOOSH and ML-TOMCAT. Also,



simulation B_ERA5 shows more significant positive trends in the tropical lowermost stratosphere during winter and spring compared to A_ERAI and ML-TOMCAT. In the NH lower stratosphere, the negative trends from A_ERAI show relatively better agreement with those from ML-TOMCAT while B_ERA5 still shows opposite and weak positive trends in most months. These seasonal trends provide more information beyond the annual mean trends, which is helpful in further understanding the role of dynamical variability in short-term trends as well as the prediction of ozone recovery.

The post-1998 seasonal ozone profile trends averaged over the three latitude bands (60-35°S, 20°S-20°N, 35-60°N) from SWOOSH, ML-TOMCAT and two simulations (A_ERAI and B_ERA5) are presented and compared in **Figure S4** with Ridge regression. The differences of the seasonal ozone profile trends using OLS and Ridge regression methods are also shown in **Figure S5**. Consistent with the monthly mean trend variations shown in **Figures 4-5**, the ozone profile trends during post-1998 time periods show seasonal and altitude dependence for all data sets. The ML-TOMCAT data set shows similar seasonal trends to those using SWOOSH data, while model simulations show larger inconsistencies especially in the lower stratosphere. The considerable differences indicate large uncertainties in estimates of seasonal ozone trends particularly in the lower stratosphere where dynamical processes dominate, and caution is required when discussing results if linear relationship (multi-collinearity) exists between the explanatory variables.

In the middle stratosphere (near 20-30 hPa), the positive trends in the SH mid-latitudes from SWOOSH data are constrained by ~2 % per decade in September-October-November (SON) with Ridge regression. Meanwhile, the negative trends in the NH mid-latitudes in JJA are also constrained by ~0.5% per decade compared to OLS regression. In the tropical lowermost stratosphere (near 100 hPa), the observed negative trends are constrained with Ridge regression by more than 2 % per decade for all seasons. For the model simulations, trends in the tropical lower stratosphere also show large differences with a wide variability for different seasons. These differences between OLS- and Ridge- based ozone profile trends imply that Ridge regression to some extent has improved the reliability of the model in the presence of multi-collinearity.

### 4.2 Ozone variations associated with natural processes

The QBO at 30 hPa and 50 hPa are important proxies used in the regression model to represent the variability of stratospheric ozone in the tropics as well as at higher latitudes (Anstey and Shepherd, 2014; Lu et al., 2019; Xie et al., 2020; Zhang et al., 2021; Wang et al., 2022). Considering the nonlinear effect, the monthly terms of QBO proxies are used for regression analyses. **Figures 6-7** show the seasonal responses of stratospheric ozone to QBO at 30 hPa and 50 hPa from SWOOSH, ML-TOMCAT and two model simulations over the long period 1984–2020 (1984-2018 for A_ERAI) based on Ridge regression. Similar results based on OLS regression are also presented in the supplementary **Figures S6-7**. It is obvious that the seasonal cycle modulates the QBO at higher latitudes with more significant responses during local winter-spring (Tung and Yang, 1994; Wang et al., 2022). A double-peaked vertical structure of stratospheric ozone anomalies associated with QBO is also clear in the tropics for all seasons. All data sets show very consistent influences of QBO on ozone, however, there exist large seasonal QBO pattern differences between various data sets when compared to the seasonal ozone trend differences shown in **Figures 4-5**. In the SH mid-latitude lower stratosphere, model simulations (A_ERAI and B_ERA5) show more negative ozone anomalies from the two QBO phases in all seasons compared to SWOOSH and ML-TOMCAT, which corresponds to the more positive ozone trends in both simulations (**Figure 5d and g**). In the tropics, the more positive ozone anomalies from model simulations near 30 hPa (**Figure 6h and k**) as well as in the lower stratosphere in DJF for both model simulations and ML-TOMCAT (**Figure 7e, h and k**), may account for the more positive ozone trends shown in **Figure 5 (b, e and h)**. The positive QBO influences on the tropical ozone and negative influences in the subtropical region are associated with the QBO phase changing from the Equator to the subtropics, which is consistent with previous studies of QBO signals in total column ozone (Tung and Yang, 1994; Chehade et al., 2014; Li et al., 2022).





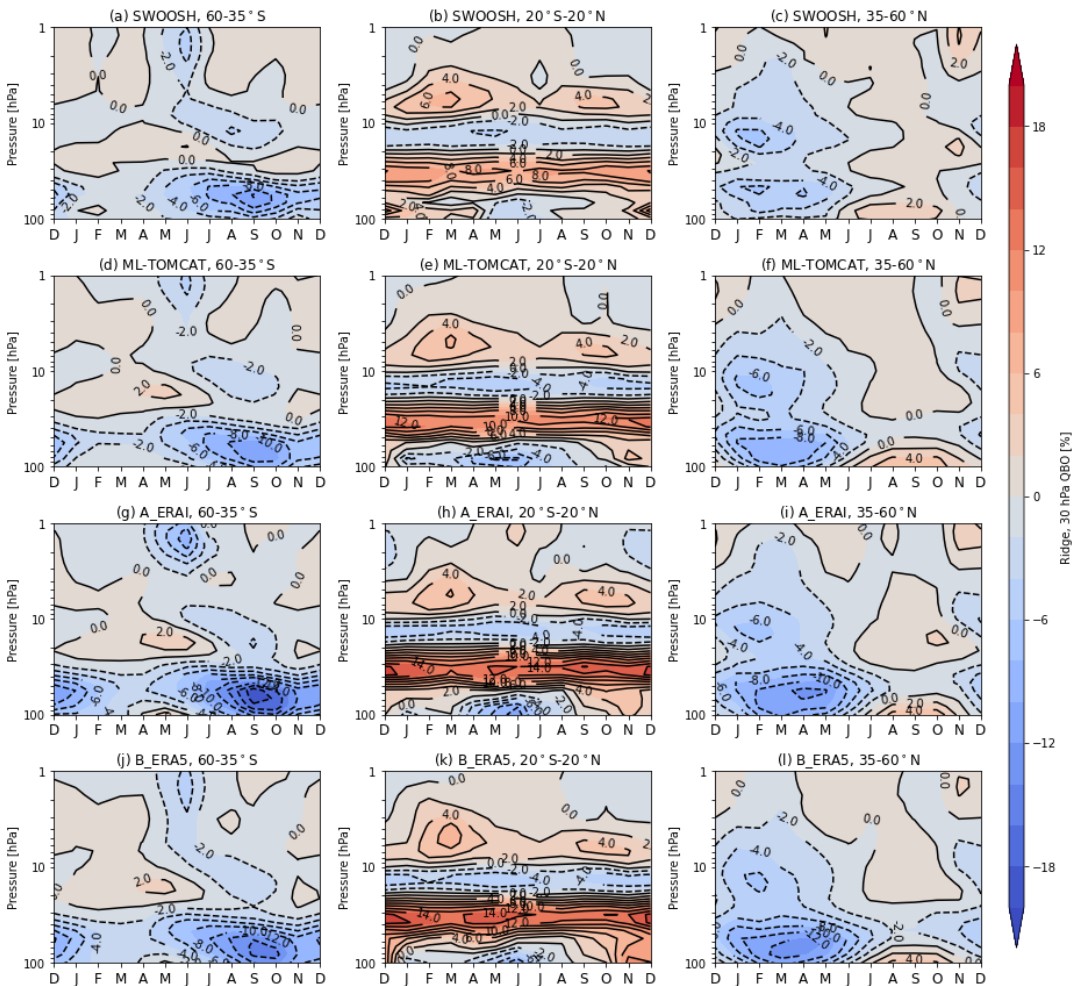

**Figure 6: Pressure-season variation of 30 hPa QBO in ozone (%) from (a-c) SWOOSH, (d-f) ML-TOMCAT, (g-i) A_ERAI and (j-l) B_ERA5 data for three selected latitudinal bands (60-35°S, 20°S-20°N, 35-60°N) based on the Ridge regression method.**


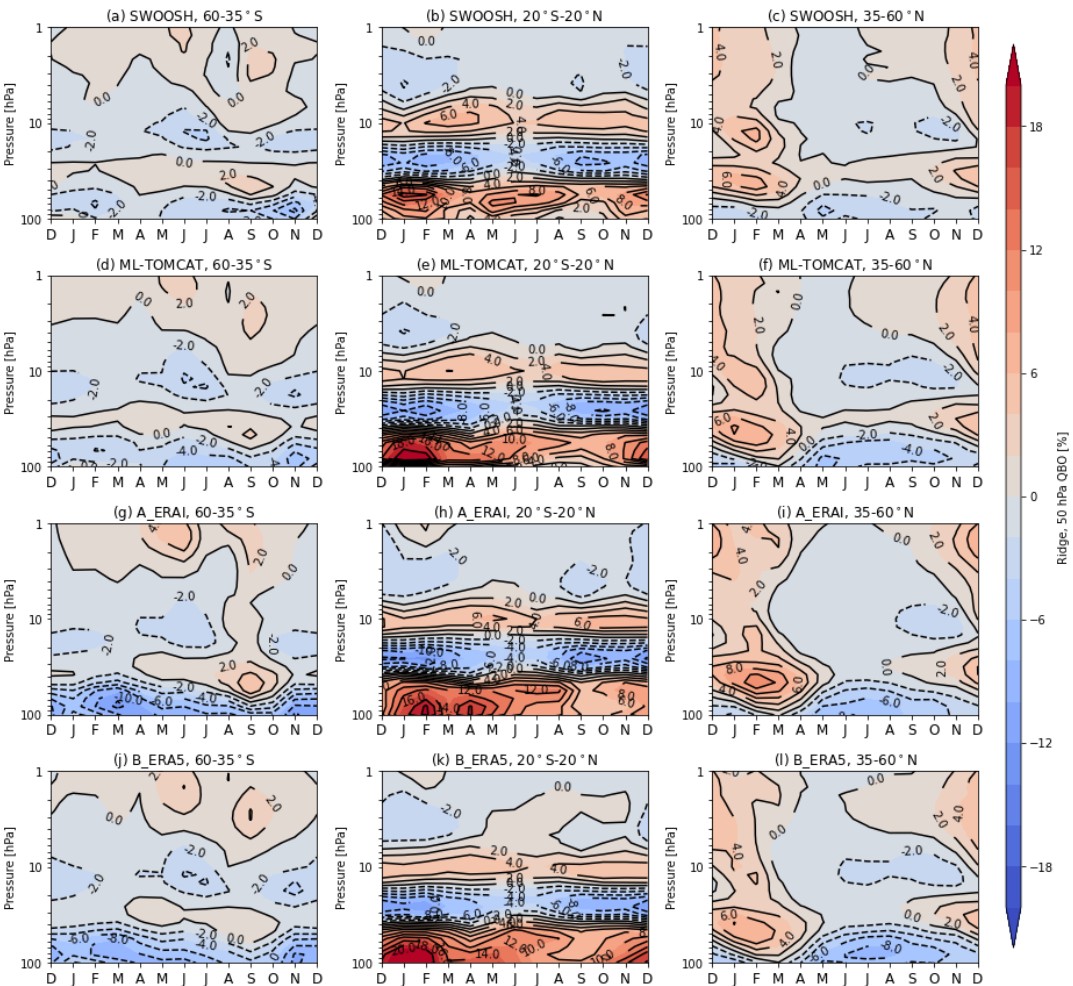

**Figure 7: Same as Figure 6 but for 50 hPa QBO in ozone (%).**

**Figure 8** shows the solar cycle response in stratospheric ozone variations derived from SWOOSH, ML-TOMCAT and two model simulations (A_ERAI and B_ERA5) based on OLS and Ridge regression methods. As expected, the coefficients of solar cycle ozone response from Ridge regression are relatively smaller than those from OLS, although both regression methods show very similar results. A common feature for all data sets is that there is a minimal solar cycle signal (negative and statistically significant) at ~10 hPa in the tropical region. This feature is similar to that in previous observations and model simulations (Soukharev and Hood, 2006; Maycock et al., 2018), but different from those with a broad single-peaked structure near 35 km (Ball et al., 2019) or 40 km (Dhomse et al., 2022). In the upper stratosphere, a U-shaped zonally averaged spatial structure is observed with maxima stretching from the tropics (3-5 hPa) to mid-latitudes (1-3 hPa), similar to that in Ball et al. (2019b) but with peaks at higher pressure levels. ML-TOMCAT data show much more consistent results with SWOOSH compared to the two model simulations. The negative solar cycle ozone response in the mid-latitude lower stratosphere confirms the dominant dynamical effect as well as very little influence from the time-varying solar fluxes (Dhomse et al., 2022). The U-shaped structure in the upper stratosphere is not well reproduced by the model simulations as the solar cycle ozone response is overestimated at most latitudes and pressure levels. Compared to simulation B_ERA5, A_ERAI shows more anomalous behaviour in the mid- and lower stratosphere, with a significant peak near 40-50 hPa in the tropics, positive solar

 

ozone response in the lower stratosphere at NH mid-latitudes, and more negative response in the southern lower stratosphere. These differences in the mid-lower stratosphere might be related to the fact that the radiative heating and subsequent changes

in Brewer-Dobson (BD) circulation are not well simulated in ERA-Interim reanalysis (e.g. Dhomse et al., 2016).

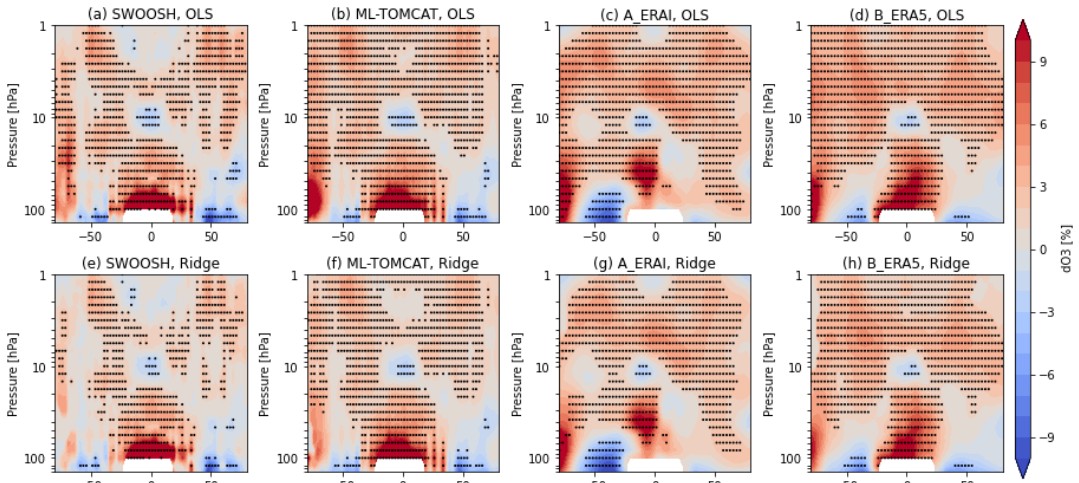

**Figure 8: Pressure-latitude cross sections of solar cycle response in the stratospheric ozone (%) derived from SWOOSH, ML-TOMCAT, and TOMCAT simulations (A_ERAI and B_ERA5) based on (a-d) OLS and (e-h) Ridge regression methods. The**
**stippling indicates regions that are significant at the 95 % level.**

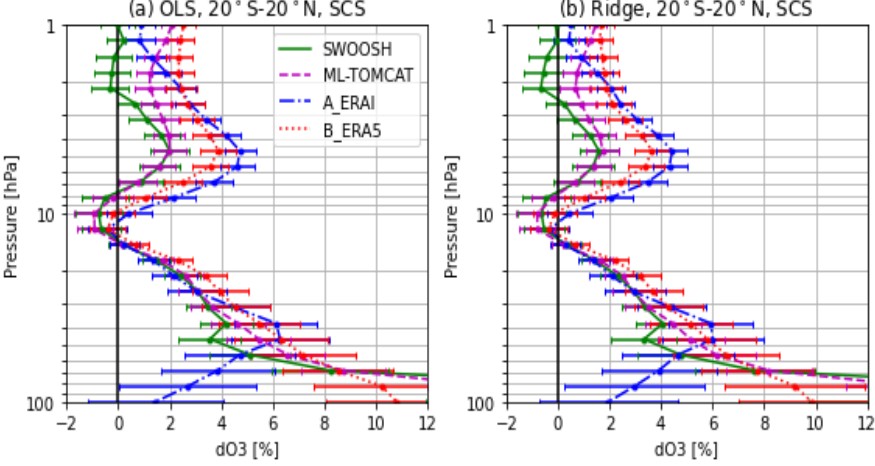

**Figure 9: Profiles of ozone solar cycle signal (SCS) for the tropical region (20°S–20°N) from SWOOSH, ML-TOMCAT as well as TOMCAT simulations A_ERAI and B_ERA5 based on (a) OLS and (b) Ridge regression methods. Error bars are 2σ uncertainties.**


**Table 3: The ozone solar cycle signals (with 2σ uncertainties) over 20°S–20°N with OLS and Ridge regression methods.**

| Levels (hPa) | SWOOSH OLS | SWOOSH Ridge | ML-TOMCAT OLS | ML-TOMCAT Ridge | A_ERAI OLS | A_ERAI Ridge | B_ERA5 OLS | B_ERA5 Ridge |
|---|---|---|---|---|---|---|---|---|
| 1 | -0.02 (0.67) | -0.09 (0.71) | 2.07 (0.39) | 1.45 (0.41) | 0.90 (0.53) | 0.50 (0.54) | 2.54 (0.45) | 1.67 (0.48) |
| 4.6 | 2.00 (0.74) | 1.58 (0.76) | 2.00 (0.61) | 1.73 (0.62) | 4.72 (0.60) | 4.42 (0.61) | 3.88 (0.67) | 3.64 (0.68) |



| 10 | -0.75 (0.90) | -0.64 (0.94) | -0.93 (0.80) | -0.81 (0.83) | 0.41 (0.87) | 0.44 (0.88) | -0.18 (0.82) | -0.14 (0.83) |
| 46.4 | 3.56 (1.21) | 3.34 (1.26) | 5.46 (1.24) | 5.14 (1.30) | 6.29 (1.93) | 6.01 (1.97) | 6.32 (1.82) | 5.79 (1.85) |
| 100 | 20.38 (8.45) | 18.70 (9.00) | 16.91 (2.52) | 16.92 (2.58) | 1.45 (2.62) | 1.95 (2.68) | 10.81 (2.76) | 9.84 (2.80) |

The solar response in tropical stratospheric ozone (20°S-20°N) is quantified and compared based on different data sets with OLS and Ridge regression methods, as shown in **Figure 9**. Regression coefficients for solar response at several pressure levels (1, 4.6, 10, 46.4 and 100 hPa) are given in **Table 3**. Generally, OLS and Ridge regression methods show very consistent results with expected slightly smaller coefficients and larger uncertainties in Ridge regression. In the upper stratosphere, all data sets show a solar cycle peak signal near 4.6 hPa, and the two model simulations overestimate the ozone solar response with peak values (~4 %) about twice larger than those (~2 %) from SWOOSH and ML-TOMCAT data. Simulation A_ERAI shows a double-peaked structure with the larger peak (~6 %) at near 46.4 hPa, while the other data sets show enhanced solar response with increased pressure levels in the lower stratosphere. Still, ML-TOMCAT data show better agreement with SWOOSH data in the tropical ozone solar response, especially in the lowermost stratosphere, when compared to TOMCAT simulations.

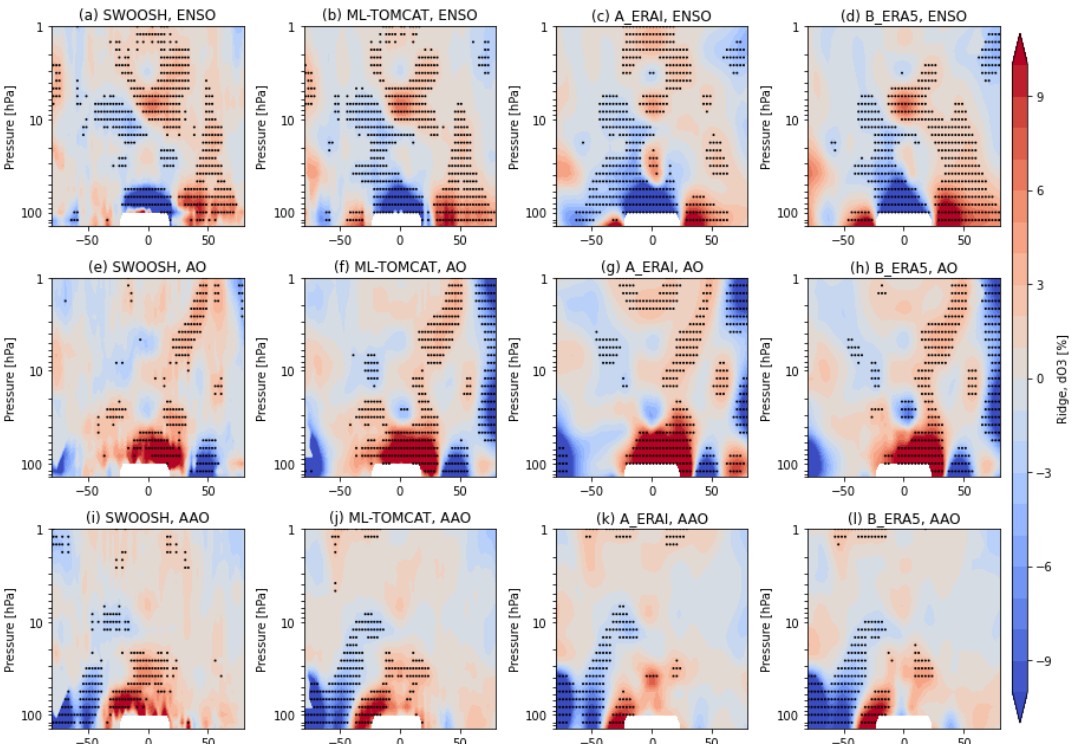

**Figure 10: Pressure-latitude cross sections of the natural ozone variations (%) associated with (a-d) ENSO, (e-h) AO and (i-l) AAO derived from SWOOSH, ML-TOMCAT, and TOMCAT simulations (A_ERAI and B_ERA5) based on the Ridge regression method. The stippling indicates regions that are significant at the 95 % level.**

In addition, ozone variations associated with natural processes (ENSO, AO and AAO) based on different data sets are shown in **Figure 10** with Ridge regression. **Figure S8** shows the results with OLS regression for comparison. Similarly, Ridge regression coefficients for ENSO, AO and AAO are slightly smaller but are consistent with those in OLS. The ENSO coefficient indicates a significant negative influence on the tropical lower stratospheric ozone, while there are positive patterns



in the northern mid-high latitudes due to enhanced transport from the tropics during warm ENSO events (Frossard et al., 2013; Rieder et al., 2013). In the southern mid-latitudes, the ENSO coefficients are statistically insignificant, implying that ENSO-related ozone variations differ by hemisphere with the ENSO phase (Ziemke et al., 2010; Oman et al., 2013). Both satellite data and model simulations capture these features, although there are still some differences. In the tropical region near 30 hPa, a significant positive ozone response to ENSO (~5 %) occurs in the simulation A_ERAI, which is different from the negligible response in all other data sets. In the lower stratosphere, simulation B_ERA5 overestimates the observed positive response in the extratropics, while A_ERAI shows significant negative responses in most regions of the SH extratropics.

The negative AO coefficients in the northern extratropics as well as the negative AAO coefficients in the southern extratropics lead to increased ozone with enhanced ozone transport (Steinbrecht et al., 2011; Chehade et al., 2014). The negative AO (AAO) indices in the extratropics are characterized by a pronounced poleward deflection of planetary waves, which means an enhanced Brewer-Dobson circulation and more ozone transport into the extratropics (Steinbrecht et al., 2011). As shown in **Figure 10**, zonally averaged ozone variations in the lower stratosphere are more sensitive to the AO and AAO indices compared to those in the middle and upper stratosphere. The regression fit has been improved by accounting for various dynamical proxies, however, these proxies are not independent and they can only partly explain the complicated structure of dynamical variability (Petropavlovskikh et al., 2019; WMO, 2022). Thus, the use of these dynamical proxies requires care especially for the lower stratospheric region.

## 5 Summary and Conclusions

In this study, we have investigated the stratospheric ozone trends and attribution with ordinary (OLS) and regularised (Ridge) multivariate regression methods. The merged satellite-based data set (SWOOSH), two TOMCAT model simulations (A_ERAI and B_ERA5) and a machine-learning-based satellite-corrected TOMCAT product (ML-TOMCAT) are used and compared over the period 1984-2020. We adopt the Ridge regression method to overcome the multi-collinearity-related issue due to the complex coupling in most atmospheric processes. We have analyzed the ozone profile trends and ozone variations associated with natural processes based on both OLS and Ridge regression methods. Our main results are summarized as follows:

- The comparison of the fitting results from OLS and Ridge regression models, as shown in Section 3 (**Figure 1** and **Figures S1-3**), indicates the differences between the two regression methods. With a penalty considered in Ridge regression, coefficients in the regression model are shrunk to a certain extent which is determined by the optimal tuning value. This optimal tuning value changes with altitude and latitude, indicating, as expected, that ozone concentrations are controlled by different processes at different altitudes and latitudes and it is inappropriate to use the same regression model for all locations. To avoid multi-collinearity-related issues, we have applied Ridge regression to quantify the stratospheric ozone trends and changes and to compare it with the conventional OLS regression method.

- We compare the stratospheric ozone profile trends for the pre- and post-1998 periods as well as the seasonal dependence with OLS and Ridge regression. Both OLS and Ridge regression methods show a strong seasonal dependence in stratospheric ozone trends. The coefficients of trend estimates at different altitudes and seasons are constrained by Ridge regression in magnitudes and fluctuations. Compared to OLS, ozone declines during 1984-1997 are smaller in Ridge regression, and largest differences between ozone trends using OLS and Ridge regression are apparent in the upper stratosphere (>1 % per decade at 2 hPa) and the lowermost stratosphere (>4 % per decade at 100 hPa) for SWOOSH data. Since 1998, upper stratospheric ozone has recovered with the largest increase of 1.21±1.15 % per decade near 2 hPa in the NH mid-latitudes, 1.15±0.97 % per decade near 2 hPa in the tropics, and 1.77±0.85 % per decade at 3.8 hPa in the SH mid-latitudes. Negative trends with large uncertainties are observed in the lower stratosphere and are most pronounced in the tropics. The largest difference between OLS and Ridge regression methods appears in the tropical lower stratosphere (with ~7 % per decade difference at 100 hPa). Comparing trend estimates from model simulations,



we find that ML-TOMCAT trends are consistent results with those using SWOOSH data. The differences between satellite-based datasets and model simulations suggest there are still large uncertainties in the lower stratosphere where 460 dynamical processes dominate.

- Ozone variations associated with natural processes such as QBO, solar variability, ENSO, AO and AAO also indicate that Ridge regression constrains the coefficients compared to the OLS-based estimates. Despite the slightly smaller coefficients and larger uncertainties in Ridge regression, there are similar characteristics in natural ozone variations for both regression methods. For example, the positive QBO influences on the tropical lower stratospheric ozone and 465 negative influences in the subtropical region are consistent with QBO signals in total column ozone. The stratospheric ozone solar cycle response shows a U-shaped spatial structure in the upper stratosphere. The enhanced transport from the tropics during warm ENSO events leads to a significant negative influence on the tropical lower stratospheric ozone and positive influences in the northern mid-high latitudes. The negative AO/AAO coefficients in the northern/southern extratropics lead to increased ozone with enhanced ozone transport. Again, ML-TOMCAT shows similar results to those 470 using SWOOSH data model simulations show larger inconsistencies especially in the lower stratosphere while.

Finally, we argue that the considerable differences between the satellite data and model simulations highlight the large uncertainties in our understanding about the lower stratospheric trends, which suggests that caution is needed while interpreting results with different methodologies and data sets.

475 *Data availability.* SWOOSH data is available from https://csl.noaa.gov/groups/csl8/swoosh/. ML-TOMCAT data is available via https://doi.org/10.5281/zenodo.5651194. The model data are available at https://doi.org/10.5281/zenodo.6988615 (Li et al., 2022, last access: Mar 2023). Climate data used in this study are available at the source and references in Sect. 2 and Sect. 3.

*Author contributions.* YL performed the data analysis and prepared the manuscript. SSD, MPC and WF performed the model 480 simulations. SSD, MPC, WF, JB, YX and DG gave support for the discussion, simulation and interpretation and helped to write the paper. All authors edited and contributed to subsequent drafts of the paper.

*Competing interests.* The authors declare that they have no conflicts of interest.

*Acknowledgements.* The modelling work is supported by National Centre for Atmospheric Science (NCAS). We thank all providers of the climate data used in this study. The model runs were performed on the Leeds ARC and UK Archer2 HPC 485 facilities.

*Financial Support.* This work was supported by the Second Tibetan Plateau Scientific Expedition and Research Program (2019QZKK0604). This work has been supported by the NERC LSO3 project (NE/V011863/1). We also acknowledge the support of the National Natural Science Foundation of China (grant no. 42192512, 2022YFF0801703) and the Natural Science Foundation for universities in Jiangsu province (grant no. 21KJB510007).

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
