# Peer review of "Quantifying stratospheric ozone trends over 1984-2020: A comparison of ordinary and regularized multivariate regression models"

_EGUsphere, 2023_

## Author Comment (AC1)

**Reply to the comments from the reviewer (Mark Weber):**

We thank Dr Weber for his useful comments and suggestions which have helped to improve the manuscript. The reviewer's comments are given below in black text, followed by our responses in blue text.

This paper reports on ozone trends derived from observations (SWOOSH dataset) and three versions of the TOMCAT chemistry-transport-model (CTM). One of the CTMs (ML-TOMCAT) has been adjusted to satellite observations, while the other two used meteorological data from different reanalyses, ERA5 and its predescessor ERA Interim (up to 2019). Two types of regression models are used for ozone trend estimates before and after the peak of stratospheric halogens occurring in the middle 1990. The first is the ordinary-least-squares regression (OLS), the second is the ridge regression. The main idea behind the ridge regression is to introduce an additional constraint in the cost function that minimises the fit coefficients. Such a regression is generally recommended to avoid overfitting. In general the ridge regression reduces the (absolute) trends (and all other fit coefficients) and on the other hand reduces the variances (and correlation) between regression model and the underlying data. The trends after 1998 in the upper stratosphere are positive and significant in agreement with other studies (~2%/decade, e.g. Godin-Beekmann et al., 2022). The ridge regression roughly halves these trends. Overall the paper is well written. Some issues still need to be addressed before acceptance of the paper.

Discussion points:
l. 30, l. 34 and other places: Differences in ozone trends at 100 hPa from OLS and ridge are larger than 4%/decade (7%/decade in the tropics), but the trend uncertainties are on the order of 24%. This means that these differences are not significantly different from zero. More relevant is the difference in the upper stratosphere (~1%/decade vs ~2%/decade), both significant. This should be mentioned here.

Reply: According to the reviewer's suggestion, we have updated our results with AR1 correction applied to the OLS regression. As mentioned by the reviewer, the trend coefficients do not change much but the uncertainties increase to some extent with this correction. Detailed figures are updated and shown in the revised manuscript.

We have modified the sentences in the abstract (Lines 29-38) based on the updated results: "For 1984-1997, we observe smaller negative trends in the SWOOSH stratospheric ozone profile using Ridge regression compared to OLS. Except for the tropical lower stratosphere, the largest differences arise in the mid-latitude lowermost stratosphere (>4% per decade difference at 100 hPa). Since 1998, and the onset of ozone recovery in the upper stratosphere, the positive trends estimated using the Ridge regression model (~1% per decade near 2 hPa) are smaller than those in OLS (~2% per decade). In the lower stratosphere, post-1998 negative trends with large uncertainties are observed and Ridge-based trend estimates are somewhat smaller and less variable in magnitude compared to the OLS regression. Aside from the tropical lower stratosphere, the largest difference is around 2% per decade at 100 hPa (with ~3% per decade uncertainties for individual trends) in northern midlatitudes. For both time periods the SWOOSH data produces large negative trends in the tropical lower stratosphere with a correspondingly large difference between the two trend methods. In both cases the Ridge method produces a smaller trend."

We also added the related information in the main text (Lines 290-295): "The largest difference between OLS and Ridge regression methods occurs in the tropical lowermost stratosphere with a difference of ~9% per decade at 100 hPa (but with larger uncertainties >10% per decade for both regression methods), followed by the NH mid-latitudes with >2% per decade difference at 100 hPa (~3% per decade uncertainties). Note that, despite the large differences between OLS and Ridge-based trends, they are still within the uncertainties of the individual trends."

l. 37: It is not surprising that ML-TOMCAT agrees better with SWOOSH than the other models. Satellite corrections are derived from the same data that are also part of SWOOSH (e.g. MLS). This should be mentioned here and also in the main text.

Reply: Yes. ML-TOMCAT agrees better with SWOOSH than the other models though it is adjusted with SWOOSH data only for the Microwave Limb Sounder (MLS) measurement period (UARS-MLS and AURA-MLS). We have added this information in the revised manuscript. For e.g., "Comparing the ML-TOMCAT-based trend estimates with the ERA5-forced model simulation, we find ML-TOMCAT shows significant improvements with much better consistency with the SWOOSH data set, despite the ML-TOMCAT training period overlapping with SWOOSH only for the Microwave Limb Sounder (MLS) measurement period." (Lines 44-46)

l. 102: A detailed comparison between ERAI-TOMCAT and ERA5-TOMCAT has been reported in Li et al. 2022. In this paper the model data have been extended to 2020, however, ERAI ends in 2019 and trends are only reported up to 2018 for the ERAI-driven model. As the differences between both models are not discussed in detail here but well covered in Li et al. 2022, it could be safely omitted from this paper.

Reply: OK. We thank the reviewer for this insightful suggestion. We have removed the ERAI-TOMCAT simulation and used only ERA5-TOMCAT to make the plots and main text clearer and easier to read. Thus, all the data sets now have the same time period (1984-2020). We also updated all the related text and figures in the revised manuscript as well as in the supplement. Modifications are marked in red in the "track-changed" version of the revised manuscript. (Note: The ERA5-TOMCAT simulation data used in the revised manuscript are updated with the same latitude bins and pressure levels as SWOOSH and ML-TOMCAT data)

l. 161: The MLR setup is very different from Li et al. 2022. For instance, now twelve (monthly) trend terms are used instead of one (annual) and more proxies are used (e.g. EP flux). Please motivate why you added more terms into the regression.

Reply: Yes, the MLR setup used here is different from Li et al. (2022). It is a modified version of that used in Dhomse et al. (2022). We use twelve (monthly) trend terms instead of one (annual) as it is better at capturing seasonal patterns, and has better sensitivity to short-term fluctuations and improved flexibility that means better goodness of fit ($R^2$). We also use more proxies (e.g. EP flux) to account for the dynamical variability of stratospheric ozone and to separate the influence of individual processes (e.g. Dhomse et al., 2022; Weber et al., 2022). Additionally, although the inclusion of the dynamical proxies will generally improve the MLR fit, the various atmospheric-dynamics-related proxies are partially correlated which makes the attribution with a MLR a little tricky. So here we focus on using Ridge regression to avoid the over-fitting issue when more proxies are added into the regression. We also added a few sentences in the revised manuscript to motivate why we added more terms into the regression (See Lines 179-182).

l. 165: Here you mention the use of the EP flux proxy, but its contribution to ozone changes is not discussed in the paper. Its contribution needs to be added in Fig. 10.

Reply: We have added the contribution from the EP flux proxy (the vertical component Fz at 50 hPa) in Figure 10 and the supplementary Figure S7. We also added discussion about its contribution to ozone changes in the revised manuscript (Lines 463-470): "Changes in the vertical component (Fz) of the stratospheric EP flux represents the ozone transport due to variations in planetary wave driving from the troposphere into the stratosphere (Fusco and Salby, 1999; Weber et al., 2003; Dhomse et al., 2006). In the tropics, the strengthened upward transport is linked to an upward shift of the maximum ozone mixing ratio in the middle stratosphere, as a result there are two cells of opposite ozone pattern near 10 hPa. A similar pattern appears at mid-latitudes due to enhanced transport by the stratospheric residual circulation. The out-of-phase between the tropics and mid-latitudes reflects the overturning Brewer-Dobson circulation (Randel et al., 2002). In the lower stratosphere, the hemispherical asymmetric ozone pattern could potentially result from the combination of changes in chemical and dynamical processes (Banerjee et al., 2016; Abalos et al., 2017)."

[Figure]

**Figure 10**: Latitude-pressure cross sections of the natural ozone variations (%) associated with (a-c) ENSO, (d-f) AO, (g-i) AAO and (j-l) EP flux (Fz50) derived from SWOOSH, ML-TOMCAT and simulation ERA5 based on the Ridge regression method. The stippling indicates regions that are significant at the 95 % level.

l. 170: Only years 1991 and 1992 have been removed to avoid the use of an aerosol proxy, but Pinatubo eruption affected more years, e.g. end of 1990, 1993 and 1994. Please comment.

Reply: We agree with the reviewer, hence we have revised the regression models. To exclude the effect from Mt. Pinatubo eruption (1991), we removed the years of data from 1991 to 1994 in the updated regression models. The updated results with two more years (1993-1994) removed show very consistent results with previous results (1991-1992), except for some minor differences (e.g. ozone trends in the tropical lower stratosphere increase slightly during 1984-1997).

l. 175: Detrending means that the long-term trends in the proxies are moved to the linear trend terms. In Weber et al. 2022 we argued that the long-term dynamic trends are largely removed by the trends in the proxies, so that linear trends are then approximating the ODS related trends. In your case, the linear trends are combining dynamic and chemical trends. That should be mentioned here.

Reply: OK. We thank the reviewer for this reminder. We have added a sentence in the revised manuscript to make it clear that the linear trends in our case are the combination of dynamic and chemical trends (See Lines 187-188): "By de-trending, the long-term trends in various proxies are moved to the linear trend terms, that is, the independent linear trends in the MLR combine both the dynamic and the ODS-related chemical trends (Weber et al., 2022)."

l. 178: Collinearity means that both vectors (or time series) are 100% correlated, which is not the case here. What you mean is that many proxies are highly correlated with each other. It is suggested to avoid the term collinear throughout the text.

Reply: We thank the reviewer for the correction and suggestion. We have checked throughout the text and revised the term "multi-collinearity" to "over-fitting/highly correlated".

l. 187: "OLS will be not robust and will result in inaccurate model." I think this is not correct. The OLS regression model will yield the same (overall) results after orthogonalising all proxies, so OLS remains robust (as also your results show). The ridge regression is another representation with different constraints, but not necessarily better than OLS. Ridge and OLS derived trends in nearly all cases agree to within the uncertainties of the trends (Figs 2 and 3). Suggest to omit this sentence.

Reply: We apologise for this incorrect statement. We have omitted this sentence in the revised manuscript. We totally agree that "The ridge regression is another representation with different constraints, but not necessarily better than OLS".

l. 202: What is the training dataset? Suggest omit "to the training data"

Reply: We have omitted "to the training data".

l. 203: Omit "when the MSE reaches the minimum"; reference to Pedregosa et al. suffices.

Reply: We have omitted "when the MSE reaches the minimum".

l. 207: "cross-valdiated MSE" needs to be explained in the text. One may also want to state the drawback of ridge regression: the fit residuals (correlation between model and regression) will be larger (smaller) than that from OLS.

Reply: We have added the explanation of the "cross-validated MSE" in the revised manuscript (Lines 215-216): "The cross-validated MSE (the average of all of the test MSEs calculated from different training and testing sets) and coefficients for the Ridge regression model are also shown as the α value grows from 0.01 to 100."

Cross-validation is a way of studying how a specific sampled data set influences the mean squared error/model fit, and provides a less sample-specific estimate of the MSE. In our case, the fit residuals (correlation between model and regression) from Ridge regression are to some extent larger (smaller) than that from OLS. The reason is probably that the original OLS regression is somewhat over-fitting and this leads to smaller errors.

l. 239: different period is used for ERAI. Does that have an effect on the trends. Shouldn't ERAI be compared with other data using the same period. see also comment earlier.

Reply: As replied earlier, we have removed ERAI-TOMCAT simulation from the manuscript and used only ERA5-TOMCAT simulation in the revised version.

l. 241: readability of numbers in the tables will be improved if only one digit is only shown, e.g. -3.4(2.5) instead of -3-39(2.47).

Reply: Thank you. To improve the readability, we have modified the numbers in the tables as well as in the main text with only one digit.

l. 249: Within the uncertainties of both regressions the trend results (ridge and OLS) are not different from each other! I think this should be mentioned in the main text as well (see earlier comment). Is the annual mean the average of the twelve monthly means? Is the uncertainty of the annual trend the standard deviation from taking the mean from the monthly values or are the uncertainties from the individual months are error-propagated into the annual mean? Please explain.

Reply: As replied earlier, we have modified the sentences in the abstract as well as in the main text. Please find them in the revised manuscript.
Yes, the annual mean is the average of the twelve monthly means, and the uncertainty of the annual trend is the standard deviation from taking the mean from the monthly values. We have added the explanation in the main text (Lines 251-253).

l. 265: mention here that the large differences in trends are within the uncertainties of the individual trends (see above).

Reply: We have modified the sentence and added the information as follows: "The largest difference between OLS and Ridge regression methods occurs in the tropical lowermost stratosphere with a difference of ~9% per decade at 100 hPa (but with larger uncertainties >10% per decade for both regression methods), followed by the NH mid-latitudes with >2% per decade difference at 100 hPa (~3% per decade uncertainties). Note that, despite the large differences between OLS and Ridge-based trends, they are still within the uncertainties of the individual trends." (Lines 290-295).

l. 335: In the lower stratosphere ridge and OLS are not reliable and fail to capture the large variability. In addition, the data quality of satellites is lower in this region. So the "linear relationship" is not the issue here

Reply: We apologise for the incorrect statement. We have modified this sentence in the revised manuscript (Lines 373-376) as follows: "The considerable differences suggest that there is a large degree of uncertainty in the estimates of seasonal ozone trends, particularly in the lower stratosphere, where dynamical processes dominate, in addition there is larger uncertainties in the satellite data. Therefore, caution is needed when discussing the results for this region, as neither regression method can reliably capture the large variability."

l. 340: "These differences between OLS- and ridge- based ozone profile trends imply that Ridge regression to some extent has improved the reliability of the model in the presence of multi-collinearity." This is not generally true as discussed above. Again: Differences between OLS and ridge-based trends are within the uncertainties of the individual trends.

Reply: We have modified this sentence as follows: "Despite these differences between OLS- and Ridge-based ozone profile trends, the even larger uncertainties e.g. in the lower stratosphere (Figure S3), indicate the ozone trends from two regression models are not different from each other." (Lines 382-384)

l. 346: "Considering the nonlinear effect, the monthly terms of QBO proxies are used for regression analyses" I do not understand what is meant to be said here. Statement can be omitted.

Reply: OK. We have omitted this statement to avoid misinterpretation.

l. 355: "corresponds to the more positive ozone trends in both simulations". To me it is not clear how long-term ozone trends can be associated with QBO (contains only periodic changes after detrending)
l. 358: "... may account for the more positive ozone trends", see previous comment

Reply: We apologize for the misleading statements. We have omitted these sentences in the revised manuscript.

l. 363: How is the anomaly defined (amplitude, i.e. max minus minimum response relative to the long term zonal mean ozone times the sign of the fit coefficient?). Please specify.

Reply: The ozone anomaly (in %) is calculated by referencing the monthly mean ozone to the climatological mean for each calendar month. As all the explanatory proxies in the regression models are normalised between 0 and 1, the contribution of the natural processes (QBO, solar, ENSO, AO, AAO and EP flux) to the percentage ozone changes can be directly denoted by the fit coefficients (also equivalent to the max minus minimum response relative to the long-term zonal mean ozone times the sign of the fit coefficient).

l. 388: ozone trends are only shown below 60degs, but solar response up to 90degs. Ozone is not well sampled above 50-60degs in the early period by SWOOSH. Is the solar response a result from a fit solely limited to the late period after 1998? Why are ozone trends above 60degs not shown?

Reply: The solar response (Fig. 8), as well as the response from other natural processes (Fig. 10), is a result from a regression fit over the whole time period 1984-2020, not solely limited to the late period after 1998. To avoid the not-well-sampled data above 50-60° in the early period by SWOOSH, we have adjusted the latitude region in Figs 8 and 10 from 80°S-80°N to 60°S-60°N, at the same time to have consistent latitude regions as shown in the ozone trend results (Figs 2-5).

l. 395: use only single digits (see earlier comments). Is the table needed as the numbers can be derived from Fig. 9?

Reply: We have checked throughout the main text and changed the numbers to one digit. As the table here is derived from Fig. 9, we removed it in the revised version.

l. 409: see comments to l. 388. Please add the results of the EP flux proxy (I guess it is the vertical component of the EP flux).

Reply: As replied earlier, we have added the results and discussion about the contribution from the EP flux proxy (the vertical component of the EP flux), as shown in the revised Figure 10 and Figure S7, and the main text (Lines 463-470)

l. 425: "The negative AO (AAO) indices in the extratropics ...". This is evident in the models and ML-TOMCAT above 60 degs but not in SWOOSH. Can this be explained? Are the regressions above 60degs problematic?

Reply: As replied earlier (l. 388), the AO/AAO response is derived from a regression fit over the whole time period 1984-2020. To avoid the not-well-sampled data above 60° by SWOOSH, we have adjusted the latitude region from 80°S-80°N to 60°S-60°N.

l. 444: "it is inappropriate to use the same regression model for all locations" Not clear what is meant here, you mean you cannot use a ridge regression with a constant tuning parameter or you mean OLS. As discussed earlier I do not think that the use of OLS is inappropriate.

Reply: What we want to say here is that for Ridge regression we cannot use a constant tuning parameter for all locations. We agree with the reviewer that the OLS regression will yield robust results when the atmospheric-dynamics-related proxies are orthogonalised (Weber et al., 2022), and the Ridge regression we use here is another representation with different constraints, but it is not necessarily better than OLS.

l. 456: "The largest difference between OLS and Ridge regression methods appears in the tropical lower stratosphere (with ~7 % per decade difference at 100 hPa).", but do not forget the trend uncertainties for both regression are very high (~23%/decade).

Reply: Yes. As replied earlier, we have revised this sentence and also checked throughout the
paper to make relevant modifications. These modifications can be found in the revised
manuscript (Lines 290-295, 506-508).
Technical (selected):
l. 37: change to "the SWOOSH dataset"    Done.
l. 58: "controlled by transport and" (omit "the")    Done.
l. 150: add Snow et al. 2014 (doi:10.1051/swsc/2014001) as reference for the MgII index
The reference has been added.
l. 183: I am not sure if "objective function" is the right term, suggest "cost function" instead.
Thanks. We have changed it to "cost function".
l. 194: "as described in Hastie" (add "as described in")    Done.
l. 204: "the Python scikrit module" (add "the")    Done.
l. 220: better: "where MSE is minimum"    Done.
l. 226: "fit residuals", I guess you mean trends    Yes. We have changed it to "trends".
l. 231: Reword: You probably mean less variability in the ridge model and lower absolute fit
coefficients in the ridge regression. Please reword.
Yes. We have modified this sentence to "Compared with the trend profiles derived from OLS
regression, the Ridge regression model has less variability and lower absolute fit
coefficients." (Lines 256-258)
l. 233: "insignificant due to large uncertainties) up to 24-24%/decade" (replace "with" with
"due to" and remove "up to")
Thanks. We have modified this sentence to "The largest ozone decreases appear in the
tropical lower stratosphere (with about -30 % per decade for OLS and -12 % per decade for
Ridge regression) although there are large uncertainties (>20 % per decade)." (Lines 259-260)
l. 233: "These large uncertainties" (remove "decreases and")    Done.
l. 239: "We note" (remove "should")    Done.
l. 249: change "compared between" to "derived from"    Done.
l. 256: "across all three" (remove "the")    Done.
l. 257: change "relatively" to "slightly"    Done.
l. 262: change "in the NH" to "at NH"    Done.
l. 280: "and ERA5 shows". remove "and" and start a new sentence here    Done.
l. 281: remove "more overestimated"    Done.
l. 289: change "monthly mean variations" to "seasonal variations"    Done.
l. 302: change "... to some extent with smaller coefficients" to "absolute ridge-based trends
and fit coefficients are smaller"    Done.
l. 310: "based on the ridge regression" (add "the")    Done.
l. 312: change "minimal" to "minimum"    Done.
l. 363: "QBO response on ozone" (add "response")    Done.
l. 373: change "there is a minimal solar cycle signal (negative and statistically significant) at
~10 hPa" to "there is a negative and statistically significant solar cycle response at ~10 hPa"
Thanks. As we have updated the results for OLS regression, we have changed this sentence
and added more information in the revised manuscript (Lines 410-439).

l. 403: "being about twice larger" (add "being")  Done.
l. 468: change "The negative AO/AAO coefficients" to "The negative phase of AO/AAO"
Done.
*PS: Some references are added according to the updated content in the revised manuscript.*

**References:**
*Abalos, M., Randel, W. J., Kinnison, D. E., and Garcia, R. R.: Using the Artificial Tracer e90*
*to Examine Present and Future UTLS Tracer Transport in WACCM, J Atmos Sci, 74,*
*3383-3403, https://doi.org/10.1175/JAS-D-17-0135.1, 2017.*
*Banerjee, A., Maycock, A. C., Archibald, A. T., Abraham, N. L., Telford, P., Braesicke, P.,*
*and Pyle, J. A.: Drivers of changes in stratospheric and tropospheric ozone between year*
*2000 and 2100, Atmos. Chem. Phys., 16, 2727–2746,*
*https://doi.org/10.5194/acp-16-2727-2016, 2016.*
*Chiodo, G., Marsh, D. R., Garcia-Herrera, R., Calvo, N., and Garcia, J. A.: On the detection*
*of the solar signal in the tropical stratosphere, Atmos. Chem. Phys., 14, 5251–5269,*
*https://doi.org/10.5194/acp-14-5251-2014, 2014.*
*Dhomse, S. S., Chipperfield, M. P., Feng, W., Hossaini, R., Mann, G. W., Santee, M. L., and*
*Weber, M.: A single-peak-structured solar cycle signal in stratospheric ozone based on*
*Microwave Limb Sounder observations and model simulations, Atmos. Chem. Phys., 22,*
*903–916, https://doi.org/10.5194/acp-22-903-2022, 2022.*
*Fusco, A. C., and M. L. Salby, Interannual variations of total ozone and their relationship to*
*variations of planetary wave activity, J. Clim., 12, 1619 – 1629,*
*https://doi.org/10.1175/1520-0442(1999)012<1619:IVOTOA>2.0.CO;2, 1999.*
*Randel, W. J., Wu, F., and Stolarski, R. S.: Changes in column ozone correlated with the*
*stratospheric EP flux, J. Meteorol. Soc. Japan, 80, 849–862,*
*https://doi.org/10.2151/jmsj.80.849, 2002.*
*Snow, M., Weber, M., Machol, J., Viereck, R., and Richard, E.: Comparison of Magnesium II*
*core-to-wing ratio observations during solar minimum 23/24, J. Space Weather Space*
*Clim., 4, A04, https://doi.org/10.1051/swsc/2014001, 2014.*
*Smith, A. and Matthes, K.: Decadal-scale periodicities in the stratosphere associated with the*
*solar cycle and the QBO, J. Geophys. Res., 113, D05311,*
*https://doi.org/10.1029/2007JD009051, 2008.*
*Weber, M., Dhomse, S., Wittrock, F., Richter, A., Sinnhuber, B. M., and Burrows, J. P.:*
*Dynamical control of NH and SH winter/spring total ozone from GOME observations in*
*1995-2002, Geophysical Research Letters, 30, 10.1029/2002gl016799, 2003.*
*Weber, M., Arosio, C., Coldewey-Egbers, M., Fioletov, V. E., Frith, S. M., Wild, J. D.,*
*Tourpali, K., Burrows, J. P., and Loyola, D.: Global total ozone recovery trends attributed*
*to ozone-depleting substance (ODS) changes derived from five merged ozone datasets,*
*Atmos. Chem. Phys., 22, 6843–6859, https://doi.org/10.5194/acp-22-6843-2022, 2022.*

**Reply to the supplementary comment from Mark Weber:**

There was one point I missed in my review. For trends from monthly mean ozone time series a correction is applied in the regression to account for autoregression (AR1). This correction does not change trends so much but increases the uncertainties due to the reduction of degree-of-freedom associated with AR. It can be applied to both OLS and Ridge regression and should be done. If not, at least a good reason should be given why it is not needed here.

Reply: We thank the reviewer for his comments and suggestions about applying a correction in the regression to account for the autoregression (AR1) for the trends from monthly mean ozone time series.

We have updated our results by including a lag-1 autocorrelation correction process in the OLS regression model with the Cochrane-Orcutt method (1949). The Cochrane-Orcutt method is a popular approach used to correct for first-order autocorrelation (AR1) in the residuals of a regression model with ordinary least squares (OLS) method (e.g. Dhomse et al., 2006; Ball et al., 2019; Petropavlovskikh et al., 2019; Bognar et al., 2022; Godin-Beekmann et al., 2022). The procedure is performed iteratively with the covariance matrix updated for each iteration until the autocorrelation coefficient has converged sufficiently (Cochrane-Orcutt, 1949; Prais and Winsten, 1954).

As mentioned by the reviewer, the trend coefficients do not change much but the uncertainties increase to some extent with this correction. It should be noted that the residuals in some region of the tropical mid-lower stratosphere are still large and auto-correlated after the AR1 correction with the Cochrane-Orcutt method. Hence, some limitations and assumptions of the Cochrane-Orcutt method should be noted, e.g.:

(1) Limited to AR1 Autocorrelation: The Cochrane-Orcutt method is specifically designed to handle first-order autocorrelation (AR1). If the autocorrelation in the residuals follows a higher-order AR process or a different pattern, this method may not be appropriate or effective.

(2) Relying on AR1 Parameter Estimation: Estimating the AR1 parameter involves making assumptions about the structure of autocorrelation and may not be reliable, especially with small sample sizes or noisy data.

(3) Parameter Interpretation: After applying the Cochrane-Orcutt correction, the estimated regression coefficients and their interpretation can be affected. The coefficients of the corrected model may not have a direct interpretation in the same way as those from the original model.

(4) Efficiency Loss: Correcting for autocorrelation may lead to a loss of statistical efficiency in parameter estimates, potentially resulting in wider confidence intervals and reduced power to detect significant effects.

(5) Diagnostics: Assessing the adequacy of the correction and the presence of any remaining autocorrelation may be challenging. Model diagnostics become essential to ensure the correction's appropriateness and to identify any model misspecification issues.

(6) Data Transformation: The method involves transforming the data and iteratively estimating parameters, which may lead to additional complexities and computational burden,
especially for large datasets.

[Figure]

**Figure RC1:** Estimates of a higher-order AR structure (AR2) of the residuals using
autocorrelation and partial autocorrelation based on SWOOSH dataset.

Figure RC1 shows a case of the AR2 structure estimated by the autocorrelation and partial
autocorrelation function of the residuals. Despite the limitations of the Cochrane-Orcutt
method, the method of the usual least squares can still yield the best linear unbiased estimates
of the regression coefficients provided the autocorrelated error terms are taken into account
(Cochrane-Orcutt, 1949).

In the Ridge regression, an additional constraint (an L2 penalty) in the cost function is
introduced to constrain the magnitudes and fluctuations of the coefficient estimates. This
constraint helps to reduce the variance of the model at the expense of no longer being
unbiased. For our current MLR setup, we choose not to apply the AR1correction to Ridge
regression. If we still apply the AR1 correction to Ridge regression as for the OLS regression,
the estimated regression coefficients can be affected as the correlation between the regression
model and underlying data becomes very poor after "correction", and the regression in this
case is in an "under-fitting" state with a very large tuning parameter. Besides, when applying
the AR1 correction to Ridge regression, the autocorrelation coefficient does not always
converge during iteration which makes it impossible to obtain the covariance matrix as in
OLS regression. Hence, care is needed when applying the AR1 correction to Ridge regression
and more detailed work can be carried out in future studies.

We have added a paragraph in the revised manuscript to clarify the differences using OLS and
Ridge regression models (Lines 231-245). In Figures RC2-3, the updated ozone trend profiles
with AR1 correction applied to the OLS regression are shown and compared with Ridge
regression results (with no AR1 correction). Please also see Figures 2-3 in the revised
manuscript.

We also updated the other figures with corrected OLS regression and more detailed modifications of the updated results are marked in red in the revised manuscript. The related code    and    data    files    are    uploaded    on    github    (https://github.com/AmyLee07/

Data-and-code-for-OLS-and-Ridge-regression.git).

[Figure]

**Figure RC2**: Profiles of annual mean stratospheric ozone trends (% per decade) compared
between OLS and Ridge regression methods for three latitude bands (60-35ºS, 20ºS-20ºN and
35-60ºN) from (a-c) SWOOSH, (d-f) ML-TOMCAT, and (g-i) ERA5 model simulation over
the period 1984-1997. Shaded regions are 2-σ uncertainties. (Data during 1991-1994 are
removed).

[Figure]

**Figure RC3:** Same as Figure RC2 but for the post-1998 time periods (1998-2020) for

SWOOSH, ML-TOMCAT and ERA5 model simulation.

**References:**

Ball, W. T., Alsing, J., Staehelin, J., Davis, S. M., Froidevaux, L., and Peter, T.: Stratospheric ozone trends for 1985–2018: sensitivity to recent large variability, Atmos. Chem. Phys.,

19, 12731–12748, https://doi.org/10.5194/acp-19-12731-2019, 2019

Bognar, K., Tegtmeier, S., Bourassa, A., Roth, C., Warnock, T., Zawada, D., and Degenstein,

D.: Stratospheric ozone trends for 1984–2021 in the SAGE II–OSIRIS–SAGE III/ISS

composite dataset, Atmos. Chem. Phys., 22, 9553–9569, https://doi.org/10.5194/acp-22-9553-2022, 2022.

Cochrane, D. and Orcutt, G. H.: Application of least squares regression to relationships containing auto-correlated error terms, J. Am. Stat. Assoc., 44, 32–61, https://doi.org/10.2307/2280349, 1949.

Dhomse, S., Weber, M., Wohltmann, I., Rex, M., and Burrows, J. P.: On the possible causes of recent increases in northern hemispheric total ozone from a statistical analysis of satellite data from 1979 to 2003, Atmospheric Chemistry and Physics, 6, 1165-1180, https://doi.org/10.5194/acp-6-1165-2006, 2006.

Godin-Beekmann, S., Azouz, N., Sofieva, V. F., Hubert, D., Petropavlovskikh, I., Effertz, P.,

Ancellet, G., Degenstein, D. A., Zawada, D., Froidevaux, L., Frith, S., Wild, J., Davis, S.,

Steinbrecht, W., Leblanc, T., Querel, R., Tourpali, K., Damadeo, R., MaillardBarras, E.,

Stübi, R., Vigouroux, C., Arosio, C., Nedoluha, G., Boyd, I., Van Malderen, R., Mahieu, E.,

Smale, D., and Sussmann, R.: Updated trends of the stratospheric ozone vertical distribution in the 60°S–60°N latitude range based on the LOTUS regression model ,

*Atmos. Chem. Phys., 22, 11657 – 11673, https://doi.org/10.5194/acp-22-11657-2022,*
*2022.*
*Prais, S. J. and Winsten, C. B.: Trend estimators and serial correlation, Cowles Commission*
*discussion paper: Statistics no. 383, 1–26, 1954.*
*Petropavlovskikh, I., Godin-Beekmann, S., Hubert, D., Damadeo, R., Hassler, B., and Sofieva,*
*V.: SPARC/IO3C/GAW Report on Long-term Ozone Trends and Uncertainties in the*
*Stratosphere, Tech. rep., SPARC, 9th assessment report of the SPARC project,*
*International Project Office at DLR-IPA, GAW Report No. 241, WCRP Report 17/2018,*
*available at: https://elib.dlr.de/126666/, 2019.*

---

## Author Comment (AC3)

**Reply to the comments from the editor (Jens-Uwe Grooß):**

Editor Review of the Manuscript "Stratospheric ozone trends and attribution over 1984-2020 using ordinary and regularised multivariate regression models" by Li et al.

As one of the reviewers of this manuscript did not commit a review and the other review was quite positive, I decided to base the decision of this manuscript on only one regular review and this editor review. Although I am not an expert on regression methods, I find the paper written clearly and understandable. Especially, the uncertainties of the derived ozone trends depending on the regression methods seem important to me. Also the depiction of the contribution of the natural processes to ozone changes is described well.

I would, however, suggest some more discussion of the results: To me it is not clear, in how far the shown differences in regression methods are now explaining the discrepancy in the lower stratosphere that was first pointed out by Ball et al. (2020). Besides the variability induced by the regression method, is there a model improvement with respect to the Multi-model-mean shown by Ball et al.? Or is this only the difference between free running CCMs and the CTM shown here. What can be learned from the machine-learning results (ML-TOMCAT)? Does the similarity with SWOOSH suggest that the basic mechanisms are well understood or would you expect this similarity as it is constructed by machine-learning using the observations? Why are the trends in the tropics so different between the two re-analyses? Is this due to the vertical velocities?

Therefore I suggest minor revisions to include this discussion, that would potentially bring the shown results better into the context of the present literature.

Reply: We thank the editor for his useful comments and suggestions about more discussion of the results, which have helped to improve the manuscript. The editor's comments are given below in black text, followed by our point-by-point responses in blue text.

(1) how far the shown differences in regression methods are now explaining the discrepancy in the lower stratosphere that was first pointed out by Ball et al. (2020).

Reply: Indeed, regression model methodologies do play some role in the trend estimation. Ball et al. (2020) used dynamical linear regression model (DLM) that attempts to determine time varying trends. Basically, DLM takes into account the temporal relationship between the dependent and independent variables, whereas OLS-type models assume that temporal relationship between dependent and independent variables is not important. So, to some extent somewhat larger negative trends shown in the study of Ball et al. (2020) are most probably due to the regression methodology adopted in their study. On the other hand, even with OLS/Ridge regression, models used here also show negative (though smaller in magnitude) in the lower stratosphere and exact causes for those trends are still not well understood. Although many recent studies (e.g., Chipperfield et al., 2018, Dietermüller et al., 2021, Li et al., 2022) attribute negative ozone trends in the lower stratosphere to dynamical changes, usage of reanalysis forcings (ERAI, ERA5, MERRA) are also not consistent, adding uncertainty in our understanding about the dynamical changes (e.g. Davis et al., 2023).

We have added some discussion about the differences in regression methods and comparison with the previous results (e.g. Ball et al., 2020) about the lower stratospheric ozone trends in the revised manuscript: "Negative trends with larger uncertainties are observed in the lower stratosphere, which are most pronounced in the tropics (-6.1$\pm$12.0 % per decade at 100 hPa), followed by the decrease at NH mid-latitudes (-1.6$\pm$3.2 % per decade at 100 hPa). The largest difference between OLS and Ridge regression methods occurs in the tropical lowermost stratosphere with a difference of ~9 % per decade at 100 hPa (but with larger uncertainties >10 % per decade for both regression methods), followed by the NH mid-latitudes with >2 % per decade difference at 100 hPa (~3 % per decade uncertainties). Note that, despite the large differences between OLS and Ridge-based trends, they are still within the uncertainties of the individual trends. The observed ozone decreases in the lower stratosphere are similar to recent records (e.g. Ball et al., 2019; 2020; Godin-Beekmann et al., 2022), which could be explained by the increased tropical upwelling and mid-latitude mixing (Wargan et al., 2018; Ball et al., 2020; Orbe et al., 2020; Davis et al., 2023). Nevertheless, the modelled lower stratospheric trends do not match those derived from observations. " (Lines 295-304)

"Compared to the trend estimates from simulation ERA5 in Figure 3, the ML-TOMCAT data set shows more consistent results with the SWOOSH data, with negative ozone trends in the tropical and NH mid-latitude lower stratosphere. Largest differences between SWOOSH- and ML-TOMCAT-based ozone trends appear in the SH mid-latitude lower stratosphere where ML-TOMCAT shows positive trends, and in the tropical mid- and lower stratosphere with close to zero trends near 60 hPa (although these trends have large uncertainties). On the other hand, trends from model simulation ERA5 show largest inconsistencies with respect to SWOOSH-based trends in the lower stratosphere. Simulation ERA5 shows positive trends for all three latitude bands that are more pronounced in the SH mid-latitudes (5.4$\pm$2.0 % per decade at 100 hPa for Ridge regression). These differences between satellite-based datasets and model simulation suggest there are still large uncertainties in the lower stratosphere where dynamical processes dominate (Dietmüller et al., 2021; Li et al., 2022). Ball et al. (2020) reported significant discrepancies in observation-model lower stratospheric ozone trends by using various satellite-based data sets and chemistry–climate models (CCMs). Although the inconsistencies vary with various datasets and fit methods (Dietmüller et al., 2021; Bognar et al., 2022), models generally do not reproduce the observations and the reason remains an open question." (Lines 305-319)

(2) Besides the variability induced by the regression method, is there a model improvement with respect to the Multi-model-mean (MMM) shown by Ball et al.? Or is this only the difference between free running CCMs and the CTM shown here.

Reply: The TOMCAT 3-D off-line chemical transport model (CTM) shown here is forced with meteorological fields from ECMWF ERA5 reanalyses (Hersbach et al., 2020) with a coherent historical assimilation of observations and full stratospheric chemistry scheme to reproduce the behaviour of ozone as closely as possible.

Objectively, there is no model improvement with respect to the Multi-model-mean (MMM) shown by Ball et al. (2020). The inconsistencies in observation-model lower stratospheric ozone trends shown in this study show some differences with those in previous study (Ball et al., 2020), which results from the difference between free-running CCMs and the CTM shown here. As replied above, we have added some discussion about the discrepancy of the ozone trends in the lower stratosphere.

(3) What can be learned from the machine-learning results (ML-TOMCAT)? Does the similarity with SWOOSH suggest that the basic mechanisms are well understood or would you expect this similarity as it is constructed by machine-learning using the observations?

Reply: The ML-TOMCAT data set we use here is a long-term chemically (and dynamically) consistent, satellite-data-based global gap-free stratospheric ozone profile data generated by applying a supervised machine-learning (ML) algorithm to the random-forest (RF) regression analysis (Dhomse et al., 2021).

The similarity or better agreement with SWOOSH is not surprising, as also mentioned by the reviewer (Dr Mark Weber), since satellite corrections used for ML-TOMCAT are derived from the same MLS data which are also part of SWOOSH, i.e. using 20 years of UARS-MLS (1991–1998) and AURA-MLS (2005–2016) measurements as a training period. However, it is also important that for the non-MLS time period, SWOOSH relies on limited (~30 profiles per day) observations from SAGE II and HALOE solar occultation instruments. So, monthly zonal means in SWOOSH data would have a limited set of observations but ML-TOMCAT would have means from all the model grid points. Dhomse et al. (2021) have demonstrated that ML-TOMCAT ozone concentrations are well within uncertainties of the observational data sets at almost all stratospheric levels, and there are significant improvements compared to the TOMCAT 3-D chemical transport model. Here we aim to illustrate that even with a limited set of denser observations to construct machine-learning based data, it still shows remarkable consistency with purely satellite measurement-based data in terms of ozone trends.

We have also added some sentences and comments about the similarity/agreement of the ML-TOMCAT data set with SWOOSH in the revised manuscript, e.g. "Comparing the ML-TOMCAT-based trend estimates with the ERA5-forced model simulation, we find ML-TOMCAT shows significant improvements with much better consistency with the SWOOSH data set, despite the ML-TOMCAT training period overlapping with SWOOSH only for the Microwave Limb Sounder (MLS) measurement period." (abstract, Lines 44-46) & "The better agreement between ML-TOMCAT and SWOOSH, due to satellite corrections derived from the same MLS measurements, show some improvements in this machine-learning based data set compared to TOMCAT chemical transport model." (Lines 306-308).

(4) Why are the trends in the tropics so different between the two re-analyses? Is this due to the vertical velocities?

Reply: Due to the differences existed between ERA5 and ERA-Interim reanalyses (e.g. vertical and horizontal resolutions, radiative transfer models and measurements assimilated, and changes in number and type of observations adopted), the differences of the trends in the tropics between the two re-analyses can be attributed to many reasons, including the different vertical velocities.

A detailed comparison between the model simulations forced by two re-analyses (ERAI and ERA5), including the difference of the stratospheric ozone profiles trends, has been reported in Li et al. (2022). From the discussion about the differences in age-of-air (AoA) tracer between two simulations, there exist some fundamental differences in the representation of Brewer-Dobson circulations between two reanalysis data sets.

As for TOMCAT setup, simulation ERA5 shows improvements in the TCO biases in the tropics compared to simulation ERAI. A possible explanation is that the finer vertical resolution in ERA5 alters vertical transport pathways that are critical for controlling ozone concentration as, within a few kilometres in the stratosphere, the ozone lifetime changes from days to a few years. Besides, simulation ERA5 shows increasing AoA trends in the whole stratosphere, while simulation ERAI shows a hemispheric dipole trend pattern with increasing AoA in the NH and decreasing trend in the SH lower stratosphere.

As differences between TOMCAT simulation forced with ERAI and ERA5 are already discussed in Li et al., (2022), the reviewer (Dr Mark Weber) suggested to omit results and discussion of ERAI. Hence, all the related comparison between the two model simulations has been removed from the revised manuscript.

**References:**

Ball, W. T., Alsing, J., Staehelin, J., Davis, S. M., Froidevaux, L., and Peter, T.: Stratospheric ozone trends for 1985–2018: sensitivity to recent large variability, Atmos. Chem. Phys., 19, 12731–12748, https://doi.org/10.5194/acp-19-12731-2019, 2019.

Ball, W. T., Chiodo, G., Abalos, M., Alsing, J., and Stenke, A.: Inconsistencies between chemistry–climate models and observed lower stratospheric ozone trends since 1998, Atmos. Chem. Phys., 20, 9737–9752, https://doi.org/10.5194/acp-20-9737-2020, 2020.

Bognar, K., Tegtmeier, S., Bourassa, A., Roth, C., Warnock, T., Zawada, D., and Degenstein, D.: Stratospheric ozone trends for 1984–2021 in the SAGE II–OSIRIS–SAGE III/ISS composite dataset, Atmos. Chem. Phys., 22, 9553–9569, https://doi.org/10.5194/acp-22-9553-2022, 2022.

Chipperfield, M. P., Dhomse, S., Hossaini, R., Feng, W., Santee, M. L., Weber, M., Burrows, J. P., Wild, J. D., Loyola, D., and Coldewey-Egbers, M.: On the cause of recent variations in lower stratospheric ozone, Geophys. Res. Lett., 45, 5718–5726, https://doi.org/10.1029/2018GL078071, 2018.

Davis, S. M., Davis, N., Portmann, R. W., Ray, E., and Rosenlof, K.: The role of tropical upwelling in explaining discrepancies between recent modeled and observed lower-stratospheric ozone trends, Atmos. Chem. Phys., 23, 3347–3361,
https://doi.org/10.5194/acp-23-3347-2023, 2023.

Dhomse, S. S., Arosio, C., Feng, W., Rozanov, A., Weber, M., and Chipperfield, M. P.:
ML-TOMCAT: machine-learning-based satellite-corrected global stratospheric ozone
profile data set from a chemical transport model, Earth Syst. Sci. Data, 13, 5711–
5729,https://doi.org/10.5194/essd-13-5711-2021, 2021.

Dietmüller, S., Garny, H., Eichinger, R., and Ball, W. T.: Analysis of recent
lower-stratospheric ozone trends in chemistry climate models, Atmos. Chem. Phys., 21,
6811–6837, https://doi.org/10.5194/acp-21-6811-2021, 2021.

Godin-Beekmann, S., Azouz, N., Sofieva, V. F., Hubert, D., Petropavlovskikh, I., Effertz, P.,
Ancellet, G., Degenstein, D. A., Zawada, D., Froidevaux, L., Frith, S., Wild, J., Davis, S.,
Steinbrecht, W., Leblanc, T., Querel, R., Tourpali, K., Damadeo, R., MaillardBarras, E.,
Stübi, R., Vigouroux, C., Arosio, C., Nedoluha, G., Boyd, I., Van Malderen, R., Mahieu, E.,
Smale, D., and Sussmann, R.: Updated trends of the stratospheric ozone vertical
distribution in the 60°S–60°N latitude range based on the LOTUS regression model ,
Atmos. Chem. Phys., 22, 11657–11673, https://doi.org/10.5194/acp-22-11657-2022,
2022.

Li, Y., Dhomse, S. S., Chipperfield, M. P., Feng, W., Chrysanthou, A., Xia, Y., and Guo, D.:
Effects of reanalysis forcing fields on ozone trends and age of air from a chemical
transport model, Atmos. Chem. Phys., 22, 10635–10656,
https://doi.org/10.5194/acp-22-10635-2022, 2022.

Orbe, C., Wargan, K., Pawson, S., and Oman, L. D.: Mechanisms Linked to Recent Ozone
Decreases in the Northern Hemisphere Lower Stratosphere, J. Geophys. Res-Atmos.,
https://doi.org/10.1029/2019JD031631, 2020.

Wargan, K., Orbe, C., Pawson, S., Ziemke, J. R., Oman, L. D., Olsen, M. A., Coy, L., and
Emma Knowland, K.: Recent decline in extratropical lower stratospheric ozone attributed
to circulation changes, Geophys. Res. Lett., 45, 5166–
5176, https://doi.org/10.1029/2018GL077406, 2018.

WMO: Scientific Assessment of Ozone Depletion: 2022, GAW Report No. 278, ISBN
978-9914-733-97-6, 2022.